# Long-term tolerance of islet allografts in nonhuman primates induced by apoptotic donor leukocytes

Amar Singh [1,11], Sabarinathan Ramachandran [1,11], Melanie L. Graham [2,11], Saeed Daneshmandi [3,11],
David Heller [1], Wilma Lucia Suarez-Pinzon[1], Appakalai N. Balamurugan[1,4], Jeffrey D. Ansite [1],
Joshua J. Wilhelm [1], Amy Yang[5], Ying Zhang [6], Nagendra P. Palani [7], Juan E. Abrahante [8],
Christopher Burlak [1], Stephen D. Miller [9], Xunrong Luo [3,5,10] & Bernhard J. Hering [1]

Immune tolerance to allografts has been pursued for decades as an important goal in transplantation. Administration of apoptotic donor splenocytes effectively induces antigen-specific tolerance to allografts in murine studies. Here we show that two peritransplant infusions of apoptotic donor leukocytes under short-term immunotherapy with antagonistic anti-CD40 antibody 2C10R4, rapamycin, soluble tumor necrosis factor receptor and anti-interleukin 6 receptor antibody induce long-term (≥1 year) tolerance to islet allografts in 5 of 5 nonsensitized, MHC class I-disparate, and one MHC class II DRB allele-matched rhesus macaques. Tolerance in our preclinical model is associated with a regulatory network, involving antigen-specific Tr1 cells exhibiting a distinct transcriptome and indirect specificity for matched MHC class II and mismatched class I peptides. Apoptotic donor leukocyte infusions warrant continued investigation as a cellular, nonchimeric and translatable method for inducing antigen-specific tolerance in transplantation.

[1] Schulze Diabetes Institute, Department of Surgery, University of Minnesota, Minneapolis, MN 55455, USA. [2] Preclinical Research Center, Department of Surgery, University of Minnesota, Minneapolis, MN 55455, USA. [3] Division of Nephrology and Hypertension, Northwestern University Feinberg School of Medicine, Chicago, IL 60611, USA. [4] Center for Cellular Transplantation, Cardiovascular Innovation Institute, Department of Surgery, University of Louisville, Louisville, KY 40202, USA. [5] Biostatistics Collaboration Center, Department of Preventive Medicine, Northwestern University Feinberg School of Medicine, Chicago, IL 60611, USA. [6] Minnesota Supercomputing Institute, University of Minnesota, Minneapolis, MN 55455, USA. [7] University of Minnesota Genomics Center, Minneapolis, MN 55455, USA. [8] University of Minnesota Informatics Institute, Minneapolis, MN 55455, USA. [9] Department of Microbiology-Immunology and Interdepartmental Immunology Center, Northwestern University, Chicago, IL 60611, USA. [10] Duke Transplant Center, Department of Medicine, Duke University School of Medicine, Durham, NC 27710, USA. [11] These authors contributed equally: Amar Singh, Sabarinathan Ramachandran, Melanie L. Graham, Saeed Daneshmandi. Correspondence and requests for materials should be addressed to S.D.M. (email: s-d-miller@northwestern.edu) or to X.L. (email: xunrong.luo@duke.edu) or to B.J.H. (email: bhering@umn.edu)

For many patients with end-stage organ failure, a transplant has become the most effective treatment option. Current immunosuppressive regimens effectively prevent acute rejection; however, their significant morbidity and their lack of efficacy in preventing chronic rejection remain serious problems. A growing population of chronically immunosuppressed transplant recipients continues to struggle with such problems, which adversely affect their survival.

Inducing tolerance to allograft would remove the need for maintenance immunosuppression and improve long-term allograft survival; yet, despite its first demonstration in small animal models >65 years ago[1] and its clinical significance, tolerance has been achieved in only a very few patients through mixed hematopoietic chimerism, which requires extensive conditioning therapy[2–4]. Likewise, in translational models in monkeys, only mixed chimerism has nearly consistently induced tolerance to same-donor kidney allografts[5].

Negative vaccination with apoptotic donor leukocytes (ADLs) represents a promising, nonchimeric strategy for inducing donor antigen-specific tolerance in transplantation[6]. Leukocytes treated ex vivo with the chemical cross-linker ethylcarbodiimide (ECDI) underwent rapid apoptosis after intravenous (IV) infusion[7]. In murine allotransplant models, IV infusions of ECDI-treated apoptotic donor splenocytes on days −7 and +1 (relative to transplant on day 0) induced robust and alloantigen-specific tolerance to minor antigen-mismatched skin grafts, to fully major histocompatibility complex (MHC)-mismatched islet allografts, and, when combined with short-term rapamycin, to heart allografts[8–10]. Most donor ECDI-treated splenocytes were quickly internalized by splenic marginal zone antigen-presenting cells (APCs), whose maturation after uptake of apoptotic bodies was arrested, resulting in selective upregulated negative, but not positive, costimulatory molecules[7,11].

After encountering recipient APCs, T cells with indirect allospecificity rapidly increased in number, followed by profound clonal contraction; the remaining T cells were sequestered in the spleen, without trafficking to allografts or allograft-draining lymph nodes[11]. Residual donor ECDI-treated splenocytes that were not internalized by host phagocytes weakly activated T cells with direct allospecificity, rendering them resistant to subsequent stimulation (anergy)[11]. ECDI-treated splenocytes also activated and increased the number of regulatory T (Treg) and myeloid-derived suppressor cells (MDSCs)[12]. Thus, in murine allotransplant models, mechanisms of graft protection induced by alloantigen delivery via ECDI-treated splenocytes involved clonal anergy of antidonor CD4+ T cells with direct specificity, clonal depletion of antidonor CD4+ T cells with indirect specificity, and regulation by CD4+ Treg cells and MDSCs[6,12,13].

In murine models of autoimmunity and allergy, IV delivery of antigens cross-linked with ECDI to the surface of syngeneic leukocytes restored antigen-specific tolerance[13,14].

Importantly, that strategy prevented both priming of naive T cells and effectively controlled responses of existing memory/effector CD4+ and CD8+ T cells[15,16]. A clinical trial involving multiple sclerosis patients affirmed the safety of IV delivery of encephalitogenic peptides after ECDI coupling to autologous leukocytes, also yielding preliminary evidence of efficacy[17].

In our study, considerably extending the findings on ECDI-treated donor splenocytes in murine allografts[10], we demonstrate stable tolerance to islet allografts in rhesus macaques (referred to as monkeys) given two ADL infusions under transient immunosuppression. We find that lasting tolerance in our model is associated with depletion of donor-specific T and B cell clones and, most prominently in recipients of one MHC class II (MHC-II) allele-matched ADL and allografts, potent and sustained regulation. Several immune cell subsets, including antigen-specific Tr1 cells[18], participate in immune regulation, suppressing post-transplant expansion of donor-reactive T cells and their recruitment to allografts.

## Results

**ADLs contract donor-specific T and B cell clones.** Monitoring cellular immunity early after ADL infusions under short-term immunosuppression in three nontransplanted, nondiabetic, one DRB-matched Cohort A monkeys (Supplementary Table 1, Fig. 1a) revealed several findings. The frequency of circulating MDSCs increased significantly, beginning 1 day after the first ADL infusion (day −6) and remained elevated throughout the end of follow-up on day +7 (Fig. 1b).

Additional studies on APC subsets in Cohort A revealed a profound downregulation of circulating HLA-DR+ monocytes from 87.73 ± 4.68% (mean ± SD) at baseline to 55.83 ± 10.69% at 3 days after the first ADL infusion (Supplementary Fig. 1a). Shortly after ADL infusions, immunosuppressed Cohort A monkeys also showed considerably lower percentages of CD80+ monocytes and dendritic cells (DCs) (Supplementary Fig. 1b, c) and increased percentages of PD-L1+ monocytes and DCs (Supplementary Fig. 1d, e).

The frequency of Ki67+CD4+ T cells increased 2.6-fold on day −5, followed by a 90% decline 3 days later and a near-total absence beginning 3 days after the second ADL infusion (Fig. 1c). The frequency of Ki67+CD8+ T cells increased 19-fold after the first ADL infusion, followed by a sharp decline beginning 4 days after the first ADL infusion and a near-total absence shortly after the second ADL infusion (Fig. 1c). After both ADL infusions, CD20+ B cells showed similar kinetics and magnitude of expansion and contraction (Fig. 1c). The frequency of interferon-gamma (IFN-γ)-secreting CD4+ T cells dropped significantly, and the frequency of interleukin (IL)-10-secreting CD4+ T cells remained unchanged (Fig. 1d). The donor-specific proliferation of CD4+ (Fig. 1e), CD8+ (Fig. 1f), and CD20+ (Fig. 1g) cells dropped significantly, whereas proliferation in response to third-party donors remained unchanged in carboxy-fluorescein diacetate succinimidyl ester-mixed lymphocyte reaction (CFSE-MLR) assays.

To track the fate of CD4+ T cells with indirect specificity for the mismatched donor MHC-I *Mamu* A004$_{27-41}$ peptide, we loaded it on the HLA DRB1*13 (the human homolog of *Mamu*-DR03) tetramer in three Cohort A monkeys. Those cells increased 5.6-fold on day −5, then declined 3.6-fold on day 0. Then 2 days after the second ADL infusion, the frequency of tetramer-positive CD4+ T cells increased 1.24-fold but significantly contracted on day 7 vs. naive monkeys (Fig. 1h, i).

Our clonotype analysis of the VDJ region in these monkeys demonstrated that the frequencies of about 30 T cell clones were altered after ADL infusions. Alterations in several T cell clones with different Vβ chains (4-Vβ5, 3 each of Vβ4, Vβ7, Vβ9, Vβ11, Vβ12, and β28; Supplementary Table 2) indicated that ADL infusions targeted multiple alloreactive clones, consistent with the notion that alloreactivity is polyclonal. Individual T cell clone analysis demonstrated abortive expansion and subsequent 5–8-fold contraction of multiple clones (Fig. 1j). Further in-depth sequencing, single-cell RNAseq, and comparative analysis with Cohort B (only immunosuppressed) are warranted to confirm that increased RNA copy number directly correlates with increase in the frequency of T cell receptor (TCR) clones. Thus several lines of evidence indicated that ADL infusions caused expansion, followed by contraction of donor-specific T and B cells.

**ADLs promote stable islet allograft tolerance in monkeys.** In 2 of the 7 streptozotocin (STZ)-diabetic one DRB-matched Cohort

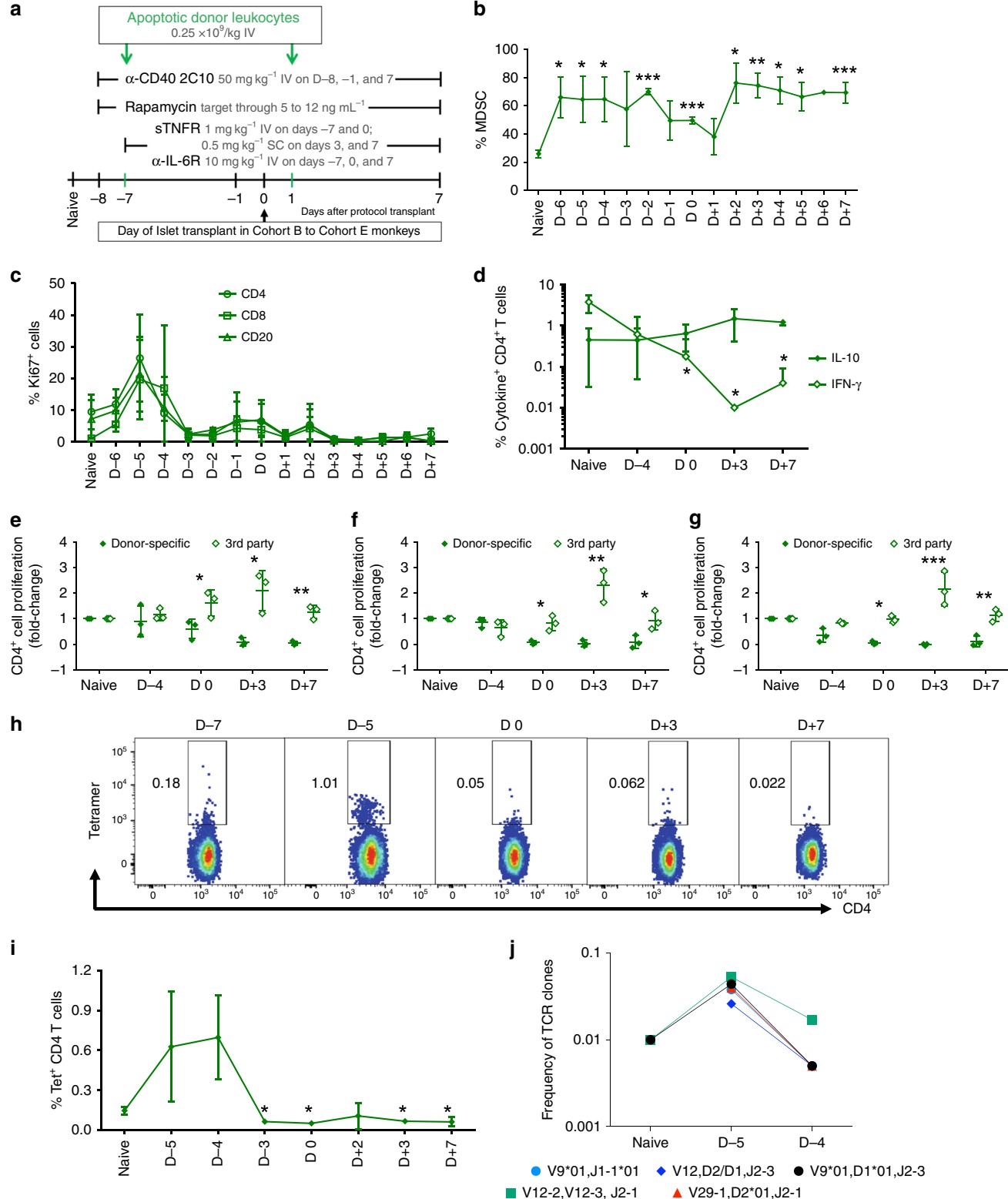

B monkeys on short-term immunosuppression, intraportal transplants of 8-day cultured islet allografts were accepted for ≥365 days (Fig. 2a, b). In 5 of the 5 Cohort C monkeys, ADL infusions added to short-term immunosuppression was associated with significantly improved survival; all 5 exhibited operational tolerance of islet allografts for ≥365 days post-transplant (Fig. 2a, b). Cohort C monkey #13EP5 became nor-moglycemic immediately posttransplant and remained so, even

after discontinuation of immunosuppression and exogenous insulin on day 21 posttransplant (Fig. 2c); that recipient's glycated hemoglobin (HbA1c) level became and remained normal post-transplant (Fig. 2d). The continued weight gain posttransplant (Fig. 2e), observed also in other Cohort C monkeys (Supplementary Table 3), is consistent with the overall safety of the treatment regimen. Pretransplant serum C-peptide levels and responses to glucose stimulation were negative in all five

**Fig. 1** Apoptotic donor leukocytes (ADLs) induce abortive expansion of donor-specific T and B cells. **a** Experimental schema describing the ADL administration protocol and the transient immunosuppression used in this Cohort A of major histocompatibility complex class I-disparate, one *Mamu*-DR matched, nondiabetic, nontransplanted monkeys (n = 3). **b** Relative frequencies of myeloid-derived suppressor cells, **c** Ki67 expression in peripheral blood lymphocytes (PBLs) of Cohort A given ADLs IV on days −7 and 0. Fluorescence-activated cell sorting analysis demonstrates peak proliferation at day −5 or day −4 followed by subsequent contraction of proliferating cells—CD4$^+$ T cells (Ki67$^+$CD4$^+$ T cells; open circle), CD8$^+$ T cells (Ki67$^+$CD8$^+$ T cells; open square), and CD20$^+$ B cells (Ki67$^+$CD20$^+$ B cells; open triangle). **d** Frequency of interferon-gamma- and interleukin-10-positive CD4$^+$ T cells collected at the indicated time points from ADL-infused Cohort A (n = 3) and restimulated in vitro with donor antigen. **e–g** Fold-change in proliferation of carboxyfluorescein diacetate succinimidyl ester-labeled recipient PBLs in response to irradiated donor (donor-specific) and third-party (3rd party) PBLs in a 6-day mixed lymphocyte reaction relative to proliferation of PBLs in naive animals (Pre-ADLs) and at the indicated time points relative to intended transplant on day 0. Data represents Mean ± SD of 3 monkeys—**e** proliferating CD4$^+$ T cells, **f** proliferating CD8$^+$ T cells, and **g** proliferating CD20$^+$ B cells. **h** Frequency of *Mamu* A*0427-41 *Mamu* DR03a tetramer$^+$ circulating CD4$^+$ T cells collected from ADL-treated Cohort A. **i** Line graphs represent the mean ± SD of 3 monkeys (n = 3). **j** Clonality of the T cells determined by high-throughput TCRβ sequencing at the individual time points. The frequency of clonal expansion was calculated by dividing the frequency of the clone at individual time points over the average frequency of all the identified mapped T cell receptor (TCR) clones. The frequencies of TCR clones at various time points are presented. Statistical analysis using paired t test was used to analyze whether a significant reduction was observed after ADL infusions when compared to naive animals. *P < 0.05; **P < 0.005; ***P < 0.0005; unpaired t test with Welch's correction. Source data are provided as a Source Data file

recipients. In monkey #13EP5, the strongly positive post-transplant fasting and random serum C-peptide levels and their increase after stimulation throughout the 1-year follow-up confirmed stable islet allograft function (Fig. 2f). That recipient showed stable posttransplant blood glucose disappearance rates (Kg) after IV challenge with glucose that were comparable with the pre-STZ rate (Fig. 2g); the C-peptide levels derived from matching tests showed substantial increases of >1 ng mL$^{-1}$ throughout the posttransplant course (Fig. 2h). Our histopathologic analysis of that recipient's liver at necropsy revealed numerous intact islets, with no or minimal periislet infiltration (Supplementary Table 4). The transplanted, intrahepatic islets showed strongly positive staining for insulin (Fig. 2i); the absence of insulin-positive islet beta cells in the native pancreas at necropsy (Fig. 2j) indicated that posttransplant normoglycemia reflected graft function and was not due to remission after STZ-induced diabetes. Cohort C monkey #15CP1 was not sacrificed at 1 year posttransplant; islet allograft function continued in that recipient for >2 years after discontinuation of immunosuppression (Supplementary Fig. 2). At necropsy of monkey #15CP1, histopathology confirmed rejection-free islet allograft survival (Supplementary Table 5) and absence of insulin-positive beta cells in the native pancreas (Supplementary Table 6). By comparison, Cohort B monkey #15CP3 became normoglycemic posttransplant but deterioration of graft function was evident starting 4 months posttransplant (Supplementary Fig. 3). Necropsy 1 month later confirmed rejection, evidenced by a small number of insulin-positive islet beta cells heavily infiltrated by mononuclear cells (Supplementary Table 7). Together, these results demonstrated the long-term functional and histologic survival of one DRB-matched islet allografts in ADL-treated monkeys, even after discontinuation of immunosuppression, indicating robust tolerance in a stringent, translational model.

**ADLs suppress effector cell expansion and function.** We compared effector cell and antibody responses in Cohort B and C recipients (Fig. 3 and Supplementary Figs. 4–8). The circulating frequency of CD3$^+$, CD4$^+$, and CD8$^+$ T cells and CD20$^+$ B cells at 3, 6, and 12 months posttransplant was not affected by ADL infusions in Cohort C (Supplementary Fig. 4a–d). However, in contrast to Cohort B monkeys not given ADLs, peritransplant ADL infusions in Cohort C were associated with prolonged suppression of expansion of circulating liver mononuclear cells (LMNCs), mesenteric lymph node (LNs), and antidonor CD4$^+$ (Fig. 3a–c) and CD8$^+$ (Fig. 3d–f) T effector memory (TEM) cells. The analysis of LMNCs and LNs was performed at the time of rejection or scheduled termination, which varied for Cohort B but

not for Cohort C animals. The percentages of CD4$^+$ and CD8$^+$ TEM cells within LMNCs and LNs were low at 1 year or later posttransplant in tolerant Cohort C monkeys as shown in Fig. 3b–e; those percentages would presumably have been equally low had the tolerant animals been sacrificed earlier before 1 year posttransplant as Cohort B monkeys that had lost graft function. Throughout the 12-month posttransplant follow-up, ADL infusions were associated with a low frequency of circulating T follicular helper (Tfh) cells in Cohort C compared with Cohort B monkeys (Fig. 3g). As with PD-1$^+$CD4$^+$ T cells (Fig. 3h), the proportion of PD-1$^+$CD8$^+$ T cells (Fig. 3i) was higher posttransplant in Cohort C vs. Cohort B, suggesting T cell-exhausted phenotype induction and elimination by ADLs. Our analyses also showed sustained suppression of Tbet$^+$CD4$^+$ and CD40$^+$CD4$^+$ T cells in the circulation of Cohort C monkeys, without affecting CD4$^+$ T cell proliferation to third-party donors (Supplementary Fig. 8a–c). The circulating frequency of Tbet$^+$CD8$^+$, CD40$^+$CD8$^+$, and CD107$^+$CD8$^+$ T cells was lower in Cohort C than in Cohort B monkeys at 3 months posttransplant, without compromised proliferation of CD8$^+$ T cells to third-party donors (Supplementary Fig. 8d–g). Our enzyme-linked immunosorbent spot (ELISPOT) analysis revealed no significant differences between Cohorts B and C in the frequency of IFN-γ-secreting T cells with direct and indirect specificities in response to irradiated donor peripheral blood lymphocytes (PBLs) at 1 month and at sacrifice, as well as no significant differences as compared with baseline (Supplementary Fig. 8h, i). Donor-specific alloimmune responses mediated through the production of multiple cytokines including IL-6 and IL-17 can mediate graft rejection in the absence of IFN-γ[19]. ADL infusions significantly suppressed IL-17 protein levels in the supernatants of donor-stimulated PBLs collected at intervals posttransplant from Cohort C recipients when compared with IL-17 levels in posttransplant Cohort B MLRs (Supplementary Fig. 8j).

The frequency of circulating CD20$^+$ B cells was similar in Cohorts B and C (Fig. 3j), but the proportion of Tbet$^+$ B cells in the circulation at 3 and 12 months posttransplant and of CD19$^+$ B cells within LMNCs at sacrifice were significantly lower in Cohort C compared with Cohort B monkeys (Supplementary Fig. 8k,l). Only Cohort B (Fig. 3k), and not Cohort C (Fig. 3l), recipients developed high donor-specific antibody (DSA) levels (expressed by mean fluorescence intensity). We also compared at an early time point posttransplant the clone sizes of donor-specific T cells in Cohorts B and C, i.e., in monkeys that received immunosuppression and donor islets without ADLs (Cohort B) and with ADLs (Cohort C). To determine donor-specific T cell expansion and depletion associated with the treatment protocols

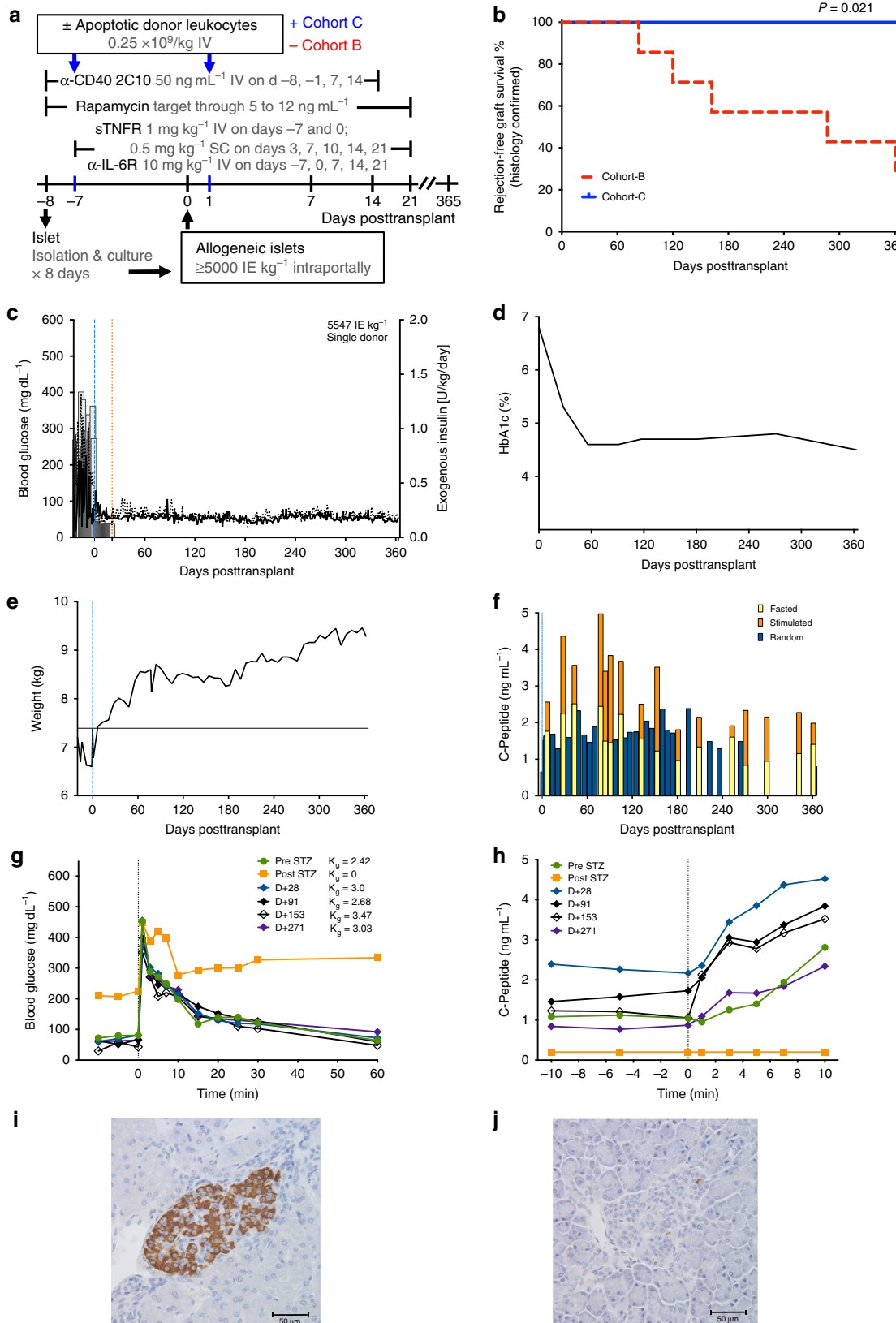

in Cohorts B and C on day 7 posttransplant, we used recipient-type MHC class II tetramers loaded with mismatched donor MHC class I peptides to compare the fold changes (vs. baseline) of the CD4+ T cells with indirect donor MHC class I specificity among the subset of circulating CD4+ T cells with non-regulatory

phenotypes. As shown in Supplementary Fig. 8m, the average fold change in 3 Cohort B monkeys was 2.55 ± 0.09, whereas there was no evidence of expansion of donor-specific, indirect CD4+ T cells in the two studied Cohort C monkeys (fold change of 0.92 ± 0.014). We did not measure DSAs frequently enough to

**Fig. 2** Apoptotic donor leukocyte (ADL) infusions facilitate stable tolerance of islet allografts. **a** Immunotherapy protocols including treatment products, dosages, routes, and timelines in Cohort B and C monkeys. sTNFR soluble tumor necrosis factor receptor (etanercept), anti-IL-6R anti-IL-6 receptor (tocilizumab), IE islet equivalent. **b** Kaplan–Meir estimates of rejection-free islet allograft survival confirmed by histology show superior sustained allograft survival in Cohort C (ADLs; $n = 5$; blue solid) compared with Cohort B (no ADLs; $n = 7$; red dashed; $P = 0.021$, Mantel–Cox). **c–j** Example of stable tolerance to islet allograft following peritransplant intravenous (IV) infusions of ADLs under the cover of transient immunosuppression (Monkey #13EP5; Cohort C). **c** Preprandial and postprandial BG (lines) and daily insulin (bars). Restoration of normoglycemia after intraportal transplant of 7-day-cultured islets (5547 IE kg$^{-1}$ by DNA). Maintenance of normoglycemia despite discontinuation of insulin and immunosuppression at day +21 posttransplant. **d** Glycated hemoglobin (HbA1c). Restoration of near-normal HbA1c levels throughout the 1-year follow-up. **e** Weight. Continued weight gain posttransplant, indicating that posttransplant euglycemia is not due to a malabsorptive state. **f** C-peptide. Positive and stable C-peptide levels (fasted, random, and mixed meal-stimulated) during the 1-year follow-up. **g** BG and Kg levels. BG before and after IV infusion of 0.5 g glucose kg$^{-1}$ (IV glucose tolerance test) and Kg levels before and after diabetes induction and 28, 91, 153, and 271 days posttransplant. Normal Kg levels posttransplant. **h** Acute C-peptide response to IV glucose (0.5 g kg$^{-1}$). Robust C-peptide increases to IV glucose during follow-up. **i** Intact, non-infiltrated, insulin-stained donor islet in recipient liver at 1 year posttransplant (a representative example of five animals studied, anti-insulin; ×40). **j** Native, insulin negative islet in pancreas at necropsy at 1 year posttransplant (a representative example of five animals studied, anti-insulin; ×40). Source data are provided as a Source Data file

determine whether DSAs were present before clinical rejection. In each of the DSA-positive recipients, rejection was confirmed by histopathologic analysis (Supplementary Table 7). Collectively, peritransplant ADL infusions impeded the posttransplant activation and expansion of effector T and B cells, as well as their recruitment to allografts in one DRB-matched monkeys on short-term immunosuppression.

**ADLs expand antigen-specific regulatory networks**. Next, we compared the frequency of lymphoid and myeloid cells with regulatory phenotypes in Cohort B and C monkeys. We found a significantly higher frequency of Tr1 cells (Supplementary Fig. 9) in the circulation at 3, 6, and 12 months posttransplant, and of LMNCs and LNs at sacrifice, in ADL-treated Cohort C than in nontreated Cohort B monkeys (Fig. 4a, b). In addition, we also found a significantly higher percentage of circulating natural suppressor (NS) and Treg cells (Fig. 4c, d) throughout the posttransplant follow-up period in ADL-treated Cohort C than in nontreated Cohort B monkeys. Regulatory B (Breg) cells (Fig. 4e, f), B10 cells (Fig. 4g, h), and MDSCs (Fig. 4i, j; Supplementary Fig. 10a) were also significantly more abundant in the circulation during the posttransplant follow-up period and, except for MDSCs, in the liver and LNs at sacrifice in Cohort C than in Cohort B monkeys.

Furthermore, additional studies on the effect of ADL infusions on circulating MDSCs on day 14 posttransplant shows a substantial increase in Cohort C (from 22.86 ± 6.20% to 47.74 ± 15.48% of CD14$^+$Lin$^-$HLA-DR$^-$ cells) and only a small increase in Cohort B (from 17.65 ± 5.80% to 24.01 ± 10.45% of CD14$^+$Lin$^-$HLA-DR$^-$ cells, Supplementary Fig. 10b). These findings extend the results on effects of ADL infusions on circulating MDSCs in Cohort A (Fig. 1b).

We also analyzed the effects of ADL infusions on APC subsets. Interestingly, when comparing Cohorts B and C, ADL infusions were associated with downregulation of HLA-DR expression in CD11b$^+$ DCs, CD14$^+$ monocytes, and only marginally in CD20$^+$ B cells at 2 and 4 weeks posttransplant, whereas HLA-DR expression increased in all three APC subsets in control Cohort B subsets (Supplementary Fig. 10c–e).

In Cohort C PBLs (as compared with unmodified recipient PBLs) at 9 and 12 months posttransplant, depletion of Treg, Breg, and Tr1 cells was associated with increased CD4$^+$ T (4.9-, 2.1-, and 8.1-fold), CD8$^+$ T (5.3-, 4.3-, and 11.1-fold), and CD20$^+$ B (3.1-, 3.0-, and 5.0-fold) cell proliferation to donor (Fig. 4k, l, Supplementary Fig. 11). Adding back Tr1 cells sorted from tolerant Cohort C recipients at 12 months posttransplant to PBL collected from recipients at baseline during re-challenge significantly suppressed donor-specific proliferation of CD4$^+$,

CD8$^+$, and CD20$^+$ cells but had no discernible effect on T and B cell proliferation in response to third-party donors (Fig. 4l).

Separation of Tr1 cells in transwell experiments did not block suppression of donor-specific responses (Fig. 4m), indicating that Tr1 cells suppressed immune responses through soluble factors. Addition of neutralizing IL-10, but not of control isotype antibody, in one-way CFSE-MLR assays significantly abrogated suppression of donor-specific responses (Fig. 4n and Supplementary Fig. 12).

Our analysis of differentially expressed genes (DEGs) in flow-sorted Tr1 cells from Cohort B and C monkeys identified 258 genes. Based on the Reactome knowledge base analysis, the genes were grouped based on the pathways impacted and are presented in Supplementary Table 8. Grouping the DEGs revealed that immune cell signaling (Supplementary Fig. 13) and mitochondrial respiration were two major biological pathways activated in Tr1 cells from Cohort C but not in Cohort B monkeys (Supplementary Table 9). Our heat map analysis of z-score of DEG demonstrated marked upregulation of immune signaling intermediates in Tr1 cells only in Cohort C (Fig. 4o). The relative transcripts of SH2D2, XBP1, and SUMO2, top three regulators in immune signaling, were significantly upregulated in Tr1 cells in Cohort C, compared to Cohort B (Fig. 4p), indicating that Cohort C Tr1 cells were in an activated state. Our heat map z-score analysis of DEG that mapped to mitochondrial respiration showed that Cohort C Tr1 cells clustered at one end, demonstrating that the cells were metabolically highly active (Fig. 4q). Members of the NDUSF family that regulate mitochondrial respiration, NDUFS4 and NDUFS5, were significantly upregulated in Cohort C Tr1 cells (Fig. 4r). Treatment of Tr1 cells sorted from a tolerant Cohort C monkeys, at 12 months posttransplant, with small interfering RNA (siRNA) targeting SH2D2 transcription molecules reduced the capacity of Tr1 cells to suppress proliferation of CD4$^+$ (59%), CD8$^+$ (53%), and CD20$^+$ (80.5%) cells in response to donor (Fig. 4s). Thus ADLs expanded regulatory networks, involving antigen-specific Tr1 cells that exhibited unique immune cell signaling and metabolic profiles.

**ADLs appear less tolerogenic in fully mismatched monkeys**. ADL infusions in fully mismatched Cohort D monkeys were associated with prolonged allograft function in two of the three recipients. But in the third recipient, the graft was rejected between 120 and 150 days posttransplant (Fig. 5a, b); in that recipient, expansion of TEM cells was not suppressed (Fig. 5c). After ADL infusions, the expansion at 6 months posttransplant of Treg (1.4-fold) and Tr1 (0.98-fold) cells in Cohort D (Fig. 5d) was less profound than in Cohort C monkeys (2.3- and 2.1-fold, Fig. 4a, d); moreover, in the Cohort D recipient whose graft was

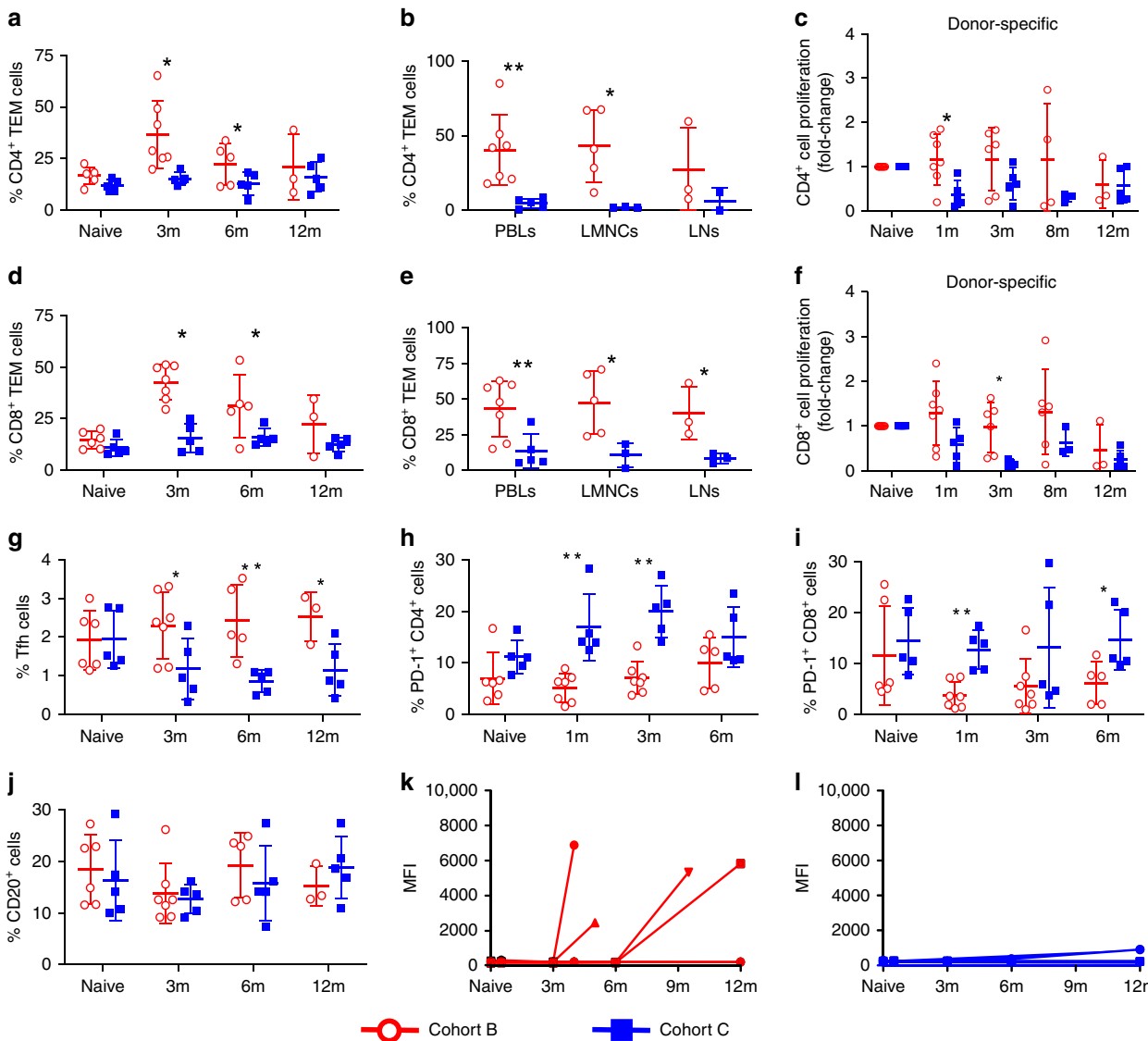

**Fig. 3** Apoptotic donor leukocyte infusions suppress effector cell expansion and function. **a** Percentage of CD4$^+$ T effector memory (TEM) cells in the peripheral blood lymphocytes (PBLs) measured longitudinally before and at 3, 6, and 12 months posttransplant in recipients from Cohorts B ($n = 7$; red) and C ($n = 5$; blue); see also Supplementary Fig. 7a, b. **b** Percentage of CD4$^+$ TEM cells in PBLs, liver mononuclear cells (LMNCs), and lymph nodes (LNs) at termination in recipients from Cohorts B ($n = 3$-7, red) and C ($n = 2$-5, blue). **c** Fold change in proliferation (compared to pretransplant levels; naive) of carboxyfluorescein diacetate succinimidyl ester (CFSE)-labeled CD4$^+$ T cells in Cohorts B and C in response to irradiated donor PBLs (donor-specific) and third-party PBLs (3rd Party; Supplementary Fig. 7c) before and at the indicated intervals posttransplant in a 6-day mixed lymphocyte reaction (MLR). **d** Percentage of circulating CD8$^+$ TEM cells in PBLs from Cohorts B ($n = 7$, red) and C ($n = 5$, blue); see also Supplementary Fig. 8d, e. **e** Percentage of CD8$^+$ TEM cells in PBLs, LMNCs, and LNs at termination in recipients from Cohorts B ($n = 3$-7, red) and C ($n = 3$-5, blue). **f** Fold change in proliferation (compared to pretransplant levels; naive) of CFSE-labeled CD8$^+$ T cells in Cohorts B and C in response to irradiated donor PBLs (donor-specific) and third-party PBL (3rd Party; Supplementary Fig. 7f) before and at the indicated intervals posttransplant in a 6-day MLR. Percentage of circulating **g** follicular helper cells (Tfh), **h** PD-1$^+$CD4$^+$ T cells, and **i** PD-1$^+$CD8$^+$ T cells. **j** Percentage of circulating CD20$^+$ B cells (see also Supplementary Fig. 7j, k). **k**-**l** Recipient IgG antibody levels against their respective donors (donor-specific antibody) expressed in mean fluorescence intensity (MFI) in recipients from **k** Cohorts B ($n = 7$, red) and **l** C ($n = 5$, blue) at the indicated time points. Unpaired $t$ test (**b**, **e**) and non-parametric Mann-Whitney $U$ test followed by post hoc analysis with the Holm-Sidak method for comparisons between two groups. (all other panels). *$P < 0.05$ and **$P < 0.01$. Source data are provided as a Source Data file

rejected, proliferation of donor-specific T cells was not suppressed (Fig. 5e, f). The frequency of three categories of Tr1 cells—IL-10, tumor growth factor-beta (TGF-β) and dual IL-10 plus TGF-β producing—significantly increased in Cohort C, but not in Cohort D, as compared with Cohort B (Supplementary Fig. 14). Tr1 cells isolated at the time of sacrifice from two Cohort D recipients with long-term allograft function reduced donor-reactive proliferation of T and B cells by >45% (Fig. 5g), as

compared with >75% for Cohort C Tr1 cells when added to CFSE-MLRs at the same ratios (Fig. 4l). Depletion of Tr1 cells from PBLs obtained at sacrifice increased donor-specific proliferation of both T and B cells ≥45% (Fig. 5g). Thus infusions of fully mismatched ADLs can also establish donor-specific regulation. More detailed studies are needed to determine whether regulation differs qualitatively or quantitatively, as compared with ADL infusions in one MHC-II-matched monkeys.

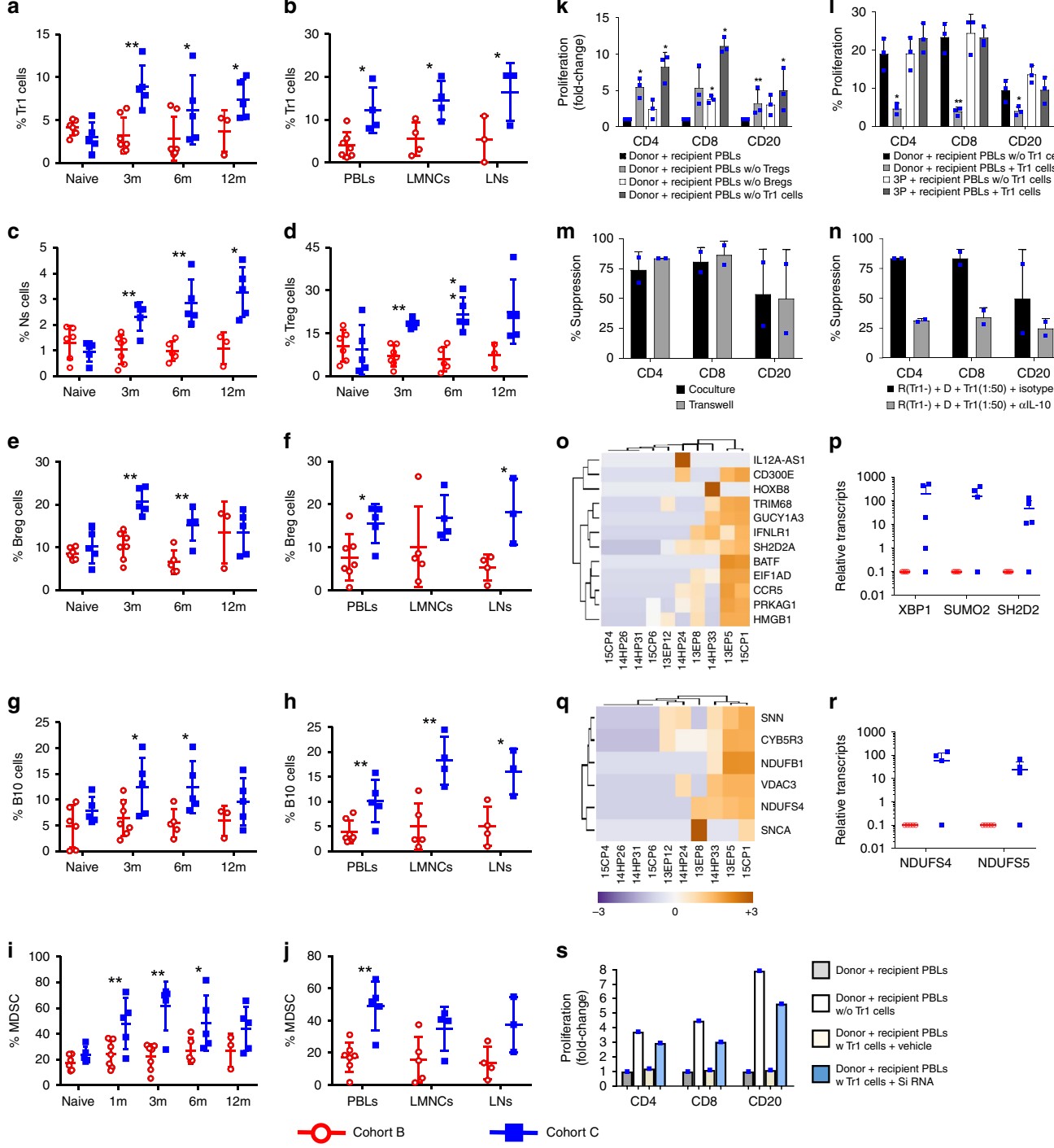

**Sensitization impedes the tolerogenic efficacy of ADLs.** In Cohort E monkey #15FP13 (not given ADLs), DSA were present at baseline but did not preclude prolonged islet allograft survival (Supplementary Table 1; Fig. 5h). In Cohort E monkey #13EP3 (given ADLs), cellular (but not humoral) memory alloreactivity was demonstrated at baseline, as was transient allograft function (Supplementary Table 1; Fig. 5i). However, in Cohort E monkeys #14HP21 and #14HP29 (both given ADLs), DSAs at baseline were associated with accelerated islet allograft loss (Fig. 5j, k). In Cohort E, posttransplant expansion of TEM cells was not suppressed (Fig. 5l) nor did circulating regulatory subsets expand even after ADL infusions (Fig. 5m). Also, donor-specific proliferation of CD4$^+$ and CD8$^+$ T cells was not inhibited (Fig. 5n,

o). Coculture of ADLs with PBLs from sensitized Cohort E monkeys caused significantly higher proliferation of donor-specific CD4$^+$ (22.4-fold), as compared with coculture of PBLs from nonsensitized Cohort C monkeys (2.8-fold, Fig. 5p). Likewise, CD8$^+$ T cells from Cohort E monkeys proliferated more strongly (23.1-fold) to ADLs in vitro, as compared with CD8$^+$ T cells from Cohort C monkeys (3.5-fold, Fig. 5p). In order to determine the functional state of the T cells, we compared the PD-1 expression on T cell subsets in PBLs in sensitized Cohort E monkeys ($n = 3$) with the PD-1 expression in nonsensitized Cohort C monkeys ($n = 5$). Interestingly, analyses of circulating T cells at days +14 and at 1 month posttransplant revealed an average 2-fold increase in the percentage of circulating PD-1

**Fig. 4** Apoptotic donor leukocyte infusions increase the frequency and function of regulatory cells. Relative numbers of circulating cells in Cohort B ($n = 7$, red) and Cohort C ($n = 5$, blue) monkeys. **a** Circulating Tr1 cells, **b** Tr1 cells in peripheral blood lymphocytes (PBLs), liver mononuclear cells (LMNCs), and lymph nodes (LNs) at the time of termination, **c** natural suppressor cells, circulating Treg (**d**), Breg and B10 cells (**e**, **g**) and in PBLs, LMNCs and LNs at the time of termination (**f**, **h**). **i** Circulating myeloid-derived suppressor cells (MDSCs) and **j** MDSCs in PBLs, LMNCs and LNs at termination. *$P < 0.05$ and **$P < 0.01$, unpaired $t$ test (**b**, **f**, **h**, **j**) and non-parametric Mann–Whitney $U$ test followed by post hoc analysis with the Holm–Sidak method for comparisons between two groups (all other panels). **k** Depletion of Tr1, Treg, and Breg cells in PBLs of Cohort C ($n = 3$) collected at 12 months posttransplant restored donor-specific proliferation of T and B cells in carboxyfluorescein diacetate succinimidyl ester-mixed lymphocyte reaction. **l** Passive transfer of flow-sorted Tr1 cells from Cohort C recipients ($n = 3$) at 12 months posttransplant abrogated the donor-specific proliferation of T and B cells in naive PBLs with no discernible effect on third-party responses. **m** Tr1 cells in a contact-independent manner regulated donor-specific immune responses and **n** addition of neutralizing anti-IL-10 antibody effectively abrogated the suppression of donor-specific proliferation of T and B cells by sorted Tr1 cells; bars represent the mean ± SD from three Cohort C monkeys. *$P < 0.05$ and **$P < 0.01$; unpaired $t$ test with Welch's correction. Heat map showing the $z$-score gene expression of **o** immune signaling intermediates and **q** metabolic pathways in Cohorts B and C monkeys. Scatter plots of transcription levels of **p** XBP1, SUMO2, and SH2D2 in PBL (at termination) and the relative expression profile of **r** NDUFS4 and NDUFS5 in PBLs (at termination) in Cohort B and C recipients. Heat map shows the differentially expressed genes with adjusted $P$ value <0.05 between the Cohort B and C monkeys. **s** RNA silencing of SH2D2 in Tr1 cell incapacitate its suppressive capacity. Fold change in donor-specific proliferation of T and B cells without Tr1 cells, Tr1cells plus vehicle, and Tr1 cells treated with small interfering RNA targeting SH2D2 transcription molecules compared to donor-treated recipient PBLs only. Source data are provided as a Source Data file

expressing CD3$^+$ (2.32 ± 0.21), CD4$^+$ (1.94 ± 0.47), and CD8$^+$ (2.12 ± 0.45) T cell subsets in Cohort C compared to Cohort E recipients (Supplementary Fig. 15a–c). These observations suggest that T cells of recipients that were sensitized to donor at baseline remained activated after ADL administration.

**One DRB-matched ADLs expand alloantigen-specific regulation.** We used MHC-II tetramers to monitor circulating CD4$^+$ T cell subsets with indirect specificities for self (shared) MHC-II and mismatched donor MHC-II and MHC-I peptides (Supplementary Fig. 16). At baseline, the frequency of CD4$^+$ T cells with indirect specificities for those peptides among subsets of CD4$^+$ T cells in Cohorts B–D varied between 1.72 ± 1.2% and 5.23 ± 3.0% (Fig. 6a–c).

As compared with baseline, the frequency of non-regulatory CD4$^+$ T cells with indirect specificity for self (shared) MHC-II peptides did not increase posttransplant in Cohorts B–D (Fig. 6d). In contrast, a sustained increase from baseline in CD4$^+$ Treg (up to 2.43 ± 0.35-fold; Fig. 6e) and Tr1 cells (up to 5.4 ± 1.2-fold; Fig. 6f) with that specificity occurred in Cohort C but not in Cohorts B and D. In Cohort D, we observed no changes posttransplant in the frequency of either non-regulatory (Fig. 6g) or regulatory (Fig. 6h, i) CD4$^+$ T cells specific for mismatched donor MHC-II peptides.

Conversely, in Cohort C, the frequency of non-regulatory CD4$^+$ T cells with specificity for mismatched donor MHC-I peptides did not change posttransplant (Fig. 6j), whereas the frequency of Treg (up to 1.93 ± 0.4-fold) and Tr1 cells (up to 3.9 ± 1.2-fold) with that specificity increased (Fig. 6k, l). The frequency of mismatched MHC-I-specific non-regulatory CD4$^+$ T cells increased posttransplant only in Cohort B (1.6 ± 0.9-fold), without any changes in the corresponding subsets of Treg and Tr1 cells. Together, ADLs expanded Treg and Tr1 cells with indirect specificities for shared (self) MHC-II and mismatched MHC-I peptides in one MHC-II matched monkeys, likely contributing to induction and maintenance of tolerance.

## Discussion

Transplantation tolerance has been pursued for decades as a clinically relevant goal[20,21]. In our study, we demonstrated that a regimen of two peritransplant ADL infusions under short-term immunotherapy safely induced long-term (>1 year) tolerance to islet allografts in five of the five nonsensitized, one MHC-II DRB allele-matched monkeys. These findings, obtained in a stringent preclinical allotransplant non-human primate (NHP) model, are

unique and point to the first clinically applicable path toward nonchimeric transplantation tolerance in humans.

Previous NHP studies reported tolerance to renal[5], but not to heart[22] or islet[23] allografts, when donor bone marrow was given under nonmyeloablative conditioning, including CD154 blockade. Of the eight monkeys so treated in one of those studies, six maintained renal allograft functions in the absence of maintenance immunosuppression for 1 year; moreover, three of them maintained long-term function without developing chronic rejection[5]. Using nonchimeric strategies in another study, tolerance of renal allografts was attained, but only inconsistently, in three of the five monkeys that received donor-specific transfusions combined with anti-CD40L for 8 weeks and rapamycin for 90 days[24]. A similar NHP strategy, as well as other previously investigated strategies, prolonged islet allograft survival after discontinuation of maintenance immunosuppression or on rapamycin monotherapy, but unlike our study, none of these protocols induced lasting tolerance[25–29]. In contrast to other cell-based tolerance strategies currently being investigated[30], our regimen did not require the adoptive transfer of regulatory cells; instead, we found that peritransplant ADL infusions under short-term immunosuppression established potent and sustained immunoregulation in vivo involving several regulatory cell types. With respect to safety, our regimen, unlike the mixed chimerism strategy, effectively induced stable tolerance without requiring irradiation, indiscriminate generalized T cell deletion, simultaneous hematopoietic stem cell transplantation, or a course of either calcineurin inhibitors or anti-CD8-depleting antibodies for control of early posttransplant direct pathway activation[5,31]. Finally, unlike other antigen-specific strategies involving soluble peptide and altered peptide ligand therapy[32,33], our ECDI-fixed leukocyte infusions were not associated with the risk of anaphylaxis[34] or with any other safety concerns in our preclinical study or in a clinical trial in multiple sclerosis[17].

Several distinct immune mechanisms were associated with our one DRB-matched ADL infusions under transient immunosuppression and tolerance to islet allografts. Our regimen depleted alloreactive effector T and B cells early after two peritransplant ADL infusions, as evidenced in Cohort A by our observation of tracking Ki67$^+$ proliferating cells, alloreactive proliferation in MLR, proliferating TCRβ clones, and CD4$^+$ T cells with indirect specificity for mismatched MHC class I allopeptides in our tetramer studies. Previous murine studies showed that uptake of apoptotic bodies by APCs following ADL infusions substantially increased PD-L1/2 expression while downregulating positive costimulation[11]. APCs exhibiting such patterns rapidly (but

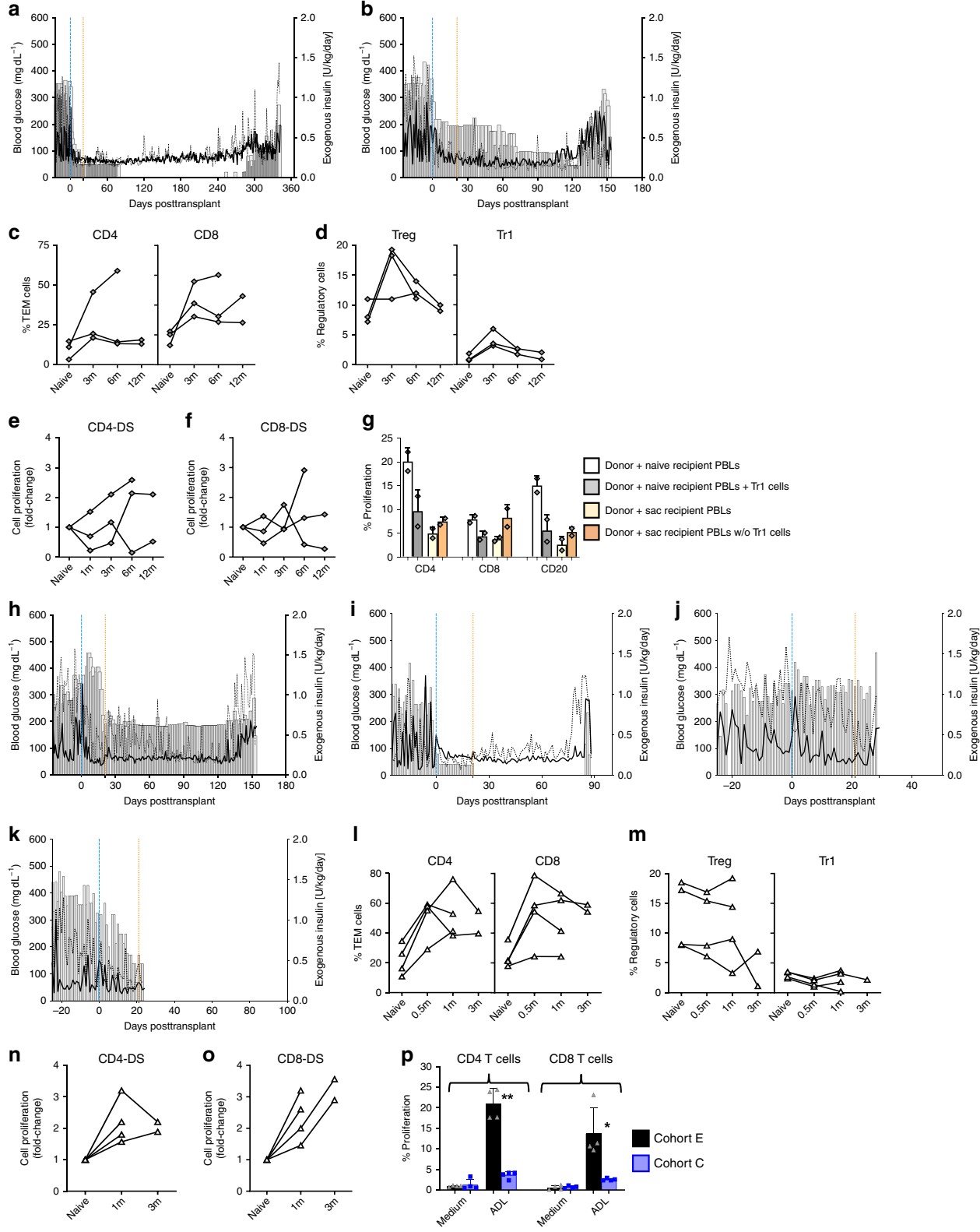

transiently) activated T cells that produce IFN-γ and IL-10 but not IL-2, IL-6, and tumor necrosis factor (TNF)-α[35,36], a cytokine microenvironment known to promote apoptotic depletion of antigen-specific T cells[37,38]. Rapamycin, part of our concomitant immunotherapy, potentiates the activation-induced cell death triggered by donor antigen under CD40:CD40L blockade[39].

In our study, we noted suppression in Cohort C (but not in Cohorts B and D) of the posttransplant expansion of circulating CD4[+] and CD8[+] TEM cells, their recruitment to the graft, and the proliferation of donor-reactive CD4[+] and CD8[+] T cells in vitro. Analysis of donor-specific proliferation showed significant differences in CD4[+] and CD8[+] T cell responses between

**Fig. 5** Graft survival and immune mechanisms in fully mismatched and sensitized monkeys. Preprandial and postprandial BG (lines) and daily insulin (bars) in **a** 15FP03 and **b** 15FP02. Fully major histocompatibility complex (MHC) mismatched monkeys (Cohort D; n = 3) poorly control T cell response and failed to sustain expanded regulatory T cells (**c–f**). Percentage of **c** CD4+ and CD8+ T effector memory (TEM) cells and **d** Treg cells and Tr1 cells. Fold change in proliferation (compared to pretransplant levels; naive) of carboxyfluorescein diacetate succinimidyl ester (CFSE)-labeled **e** CD4+ T cells and **f** CD8+ T cells in response to donor peripheral blood lymphocytes (PBLs) (donor-specific (DS)) and third-party PBL (3P) at a given time point. **g** Donor-specific suppression of T and B cells proliferation by Tr1 cells. Relative number of proliferating T and B cells when flow-sorted Tr1 cells from fully MHC-mismatched monkeys (Cohort D; n = 2) at the time of sacrifice (sac) added back to donor-stimulated naive recipient PBLs or depleted from PBLs obtained from the time of sacrifice. **h–k** Preprandial and postprandial BG (lines) and daily insulin (bars) in four sensitized islet allograft monkeys from Cohort E. **h** 15FP13, **i** 13EP3, **j** 14HP21, and **k** 14HP29. Sensitized islet allograft monkeys (Cohort E; n = 4) demonstrated expanded circulatory frequency of **l** CD4+ and CD8+ TEM cells. **m** Poor expansion of Treg cells and Tr1 cells. **n–p** Fold-change proliferation (compared to pretransplant levels; naive) of CFSE-labeled **n** CD4+ T cells and **o** CD8+ T cells in response to donor PBLs (DS) and third-party PBL (3P) at a given time point. **p** Comparative CD4+ and CD8+ T cell proliferative responses in Cohorts E (n = 4) and C (n = 4) recipients with and without apoptotic donor leukocytes. Data bars represent the mean ± SD. Unpaired t test with Welch's correction. *P < 0.05 and **P < 0.01. Source data are provided as a Source Data file

Cohorts B and C at 1 month and 3 months posttransplant (Fig. 3c, f). Though IL-17 levels in posttransplant MLR cultures were suppressed in Cohort C but not in Cohort B, we did not see a significant difference between the Cohorts in the frequency of IFN-γ-secreting T cells in response to irradiated donor PBLs in ELISPOT assays. Several studies have demonstrated that donor-specific alloimmune responses can be mediated through the production of cytokines other than IFN-γ including IL-6[40,41] and IL-17[42–44] and that CD4+ T cells with indirect specificity can mediate skin graft rejection in the absence of IFN-γ[19,45]. These findings suggest that our ADL infusions and one DRB matching did play important roles in tolerance induction and maintenance. The restored T cell proliferation to donor that we observed in vitro after depletion of regulatory subsets suggests that donor-specific T cell clones were neither deleted nor anergized but rather that regulation controlled their posttransplant expansion and effector function.

Further supporting that interpretation, we showed that the addition of ADL infusions to short-term immunosuppression in Cohort C established a regulatory network characterized by significant and sustained increases in circulating MDSCs and Tr1, Treg, NS, Breg, and B10 cells. At termination, Tr1 cells were also significantly more prevalent within livers bearing allografts and in lymph nodes of Cohort C (vs. Cohort B) recipients. It is now well established that Tr1 cell generation is mediated by APCs. Tr1 cells are induced in the periphery from naive CD4+ T cells upon TCR stimulation by APCs under tolerogenic conditions in an IL-10-enriched microenvironment, with distinct subsets of APCs, e.g., DC-10 cells, being the major source of IL-10[46–48]. Previous studies demonstrated IL-10 production triggered by apoptotic debris[49] and rapid and sustained IL-10 release from splenic marginal zone APCs after their uptake of IV infused, ECDI-treated, apoptotic leukocytes[7]. ADL infusions also altered the phenotype of APCs in our study, suggestive of their possible involvement in Tr1 cell induction and expansion. The detailed mechanisms underlying the formation of that regulatory network remain to be defined.

Nonetheless, it is possible that our one DRB-matched ADL infusions provided copious amounts of shared MHC-II peptides for presentation by MHC-II molecules on host spleen marginal zone APCs and on host liver sinusoidal endothelial cells[11,50–52]. It is known that, after trogocytosis to activated T cells[53], such peptide MHC-II complexes can deliver potent activation signals to thymus-derived Treg (tTreg) cells[54,55], which have a TCR repertoire skewed toward self-recognition[56]. Treg cells are known to promote the generation of IL-10-producing Tr1 cells[57], but it remains to be determined whether the expansion of Tr1 cells in our study was due to the influence of activated tTregs and resulted from de novo formation and/or conversion of donor-reactive T effector cells[18]. In autoimmunity models, Tr1-like cells, generated by nanoparticles coated with

autoimmune disease-relevant peptides bound to self MHC-II, are known to contribute to regulatory network formation by driving the differentiation of cognate B cells into disease-suppressing regulatory B cells[58]. Several lines of evidence suggest that CD24hiCD38hi Breg cells and CD24hiCD27+ B10 cells are IL-10-producing B cells with potent regulatory function in autoimmunity, infection, and transplantation[59–61]. Our ex vivo data suggest that Breg cells suppress the proliferation of donor-specific T cells. Nevertheless, further studies are needed to verify the regulatory function of individual regulatory subsets expanded by ADL infusions as well as their interaction. Consistent with the idea that matched MHC-II peptides facilitated regulatory networks in our study, the frequency of circulating Treg, Tr1, and Breg cells in Cohort C recipients of one DRB-matched ADLs was significantly higher than the frequency in fully mismatched Cohort D recipients.

Among regulatory subsets, we found that Tr1 cells exhibited the most potent suppression of donor-specific proliferation of T and B cells, which was mediated in part through IL-10. In contrast, third-party responses were not affected by sorted Tr1 cells, indicating their antigen specificity. In Cohort C (but not in Cohorts B and D), our tetramer studies revealed sustained posttransplant increases in circulating Treg and Tr1 cells with indirect specificity for mismatched donor MHC-I peptides. That finding corroborated their antigen specificity and was consistent with previous studies of murine and human allograft recipients showing regulation induced by mismatched MHC-I peptide presentation by shared self MHC-II molecules after one MHC-II allele-matched blood transfusions[62–64]. ADLs increased regulatory subsets in fully mismatched murine allograft recipients[6]; the effect of MHC-II matching in these models remains to be studied. In Cohort C recipients, Tr1 cells exhibited unique immune cell signaling, including significantly increased levels of SH2D2. T cell-specific adapter protein, the gene product of Sh2D2a, regulates TCR signaling through its interaction with Lck[65]; however, its absence promotes systemic autoimmunity[66]. In Cohort C, Tr1 cell transcriptomic profiles also demonstrated increased mitochondrial respiratory activity and energy utilization in Tr1 cells, revealing their activated state[67]. Whether tolerance was sustained in our study by splenic Tr1 cells remains to be addressed[68].

In Cohort B (no ADL infusions), two of the seven recipients maintained immunosuppression-free allograft survival for 1 year posttransplant, and all seven avoided acute rejection, confirming that favorable allograft survival can be achieved when early direct pathway activation is suppressed with potent induction in MHC-II-matched recipients[69]. However, this regimen failed to control indirect pathway activation, as evidenced by de novo DSA development in most Cohort B recipients[70]. The tolerogenic efficacy of one DRB-matched ADL infusions under transient immunosuppression was limited in sensitized Cohort E

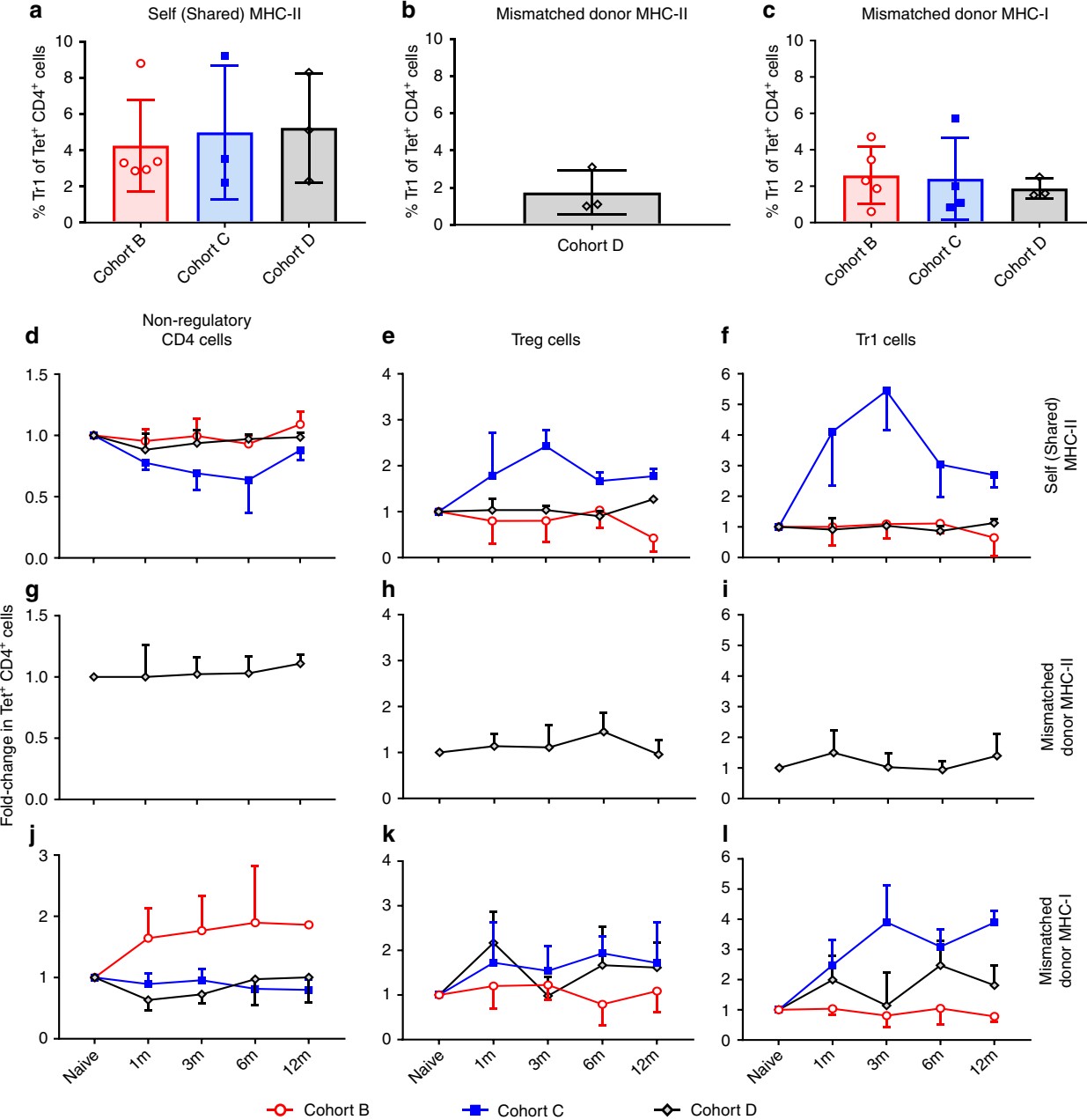

**Fig. 6** Tracking of antigen-specific CD4+ T cells. Tracking of non-regulatory CD4+ T cells, Tr1, and Treg cells with indirect specificity for self (shared) major histocompatibility complex (MHC)-II, mismatched donor MHC-II, and mismatched donor MHC-I peptides among Cohort B ($n = 5$), C ($n = 3$ or 4), and D ($n = 3$) monkeys. **a**–**c** At baseline, the percentage of Tr1 cells within tetramers + CD4+ T cells with indirect specificities for **a** self (shared) MHC II, **b** mismatched donor MHC-II and **c** mismatched donor MHC-I in Cohorts B–D (Supplementary Fig. 14, Online Methods). **d**–**f** Compared with baseline, fold change in frequency of **d** non-regulatory CD4+ T cells, **e** Treg cells, and **f** Tr1 cells with indirect specificity for self (shared) MHC-II peptide among Cohorts B–D. **g**–**i** No substantial fold changes in tetramer+ **g** non-regulatory CD4+ T cells or regulatory **h** Treg cells and **i** Tr1 cells frequency specific for mismatched donor MHC-II peptide in Cohort D. **j**–**l** Compared with baseline, fold-change frequency of **j** non-regulatory CD4+ T cells, **k** Treg cells, and **l** Tr1 cells with indirect specificity for mismatched donor MHC-I peptide among cohorts B–D. Source data are provided as a Source Data file

recipients, particularly in those with preformed DSAs, in whom memory T cells, including those not secreting IFN-γ, are mandatory for DSA responses[71,72], and in whom APCs, activated by uptake of DSA-opsonized ADLs, likely primed instead of tolerized donor-reactive T cells[73].

In summary, our study suggests that the long-pursued goal of transplantation tolerance is attainable with a nonchimeric ADL strategy that establishes a sustained and antigen-specific regulatory network. Our findings could have important implications for achieving clinical transplantation tolerance.

## Methods

**Study animals**. The cohorts included purpose-bred monkey (*Macaca mulatta*) donors and recipients of Indian origin obtained from the National Institute of Health and Infectious Diseases colony at AlphaGenesis, Inc, Yemassee, SC. Demographics of the recipient monkeys are presented in Supplementary Table 1. The exploratory group included 3 males aged 7.3 ± 0.1 years and weighed 12.5 ± 1.5 kg. The control cohort included 8 males aged 4.3 ± 2.1 years and weighed 6.2 ± 1.6 kg. Experimental cohort included 7 males and 1 female aged 4.1 ± 1.7 years and weighed 5.2 ± 1.2 kg. The donor cohort included 19 males aged 6.7 ± 3.3 years and weighed 11.7 ± 3.6 kg.

Animals were free of herpes virus-1 (B virus), simian immunodeficiency virus, type D simian retrovirus, and simian T-lymphotropic virus. Eligibility additionally

included ABO compatibility and study-defined MHC matching (MHC-I-disparate and one MHC-II DRB allele-matched donor–recipient pairs; Supplementary Table 10–14). All animals underwent high-resolution MHC-I and -II genotyping by 454 pyrosequencing (Genetics Services Unit at the Wisconsin National Primate Research Center). They had free access to water and were fed biscuits (Harlan Primate Diet 2055C, Harlan Teklad, Madison, WI) based on body weight (BW). Their diet was enriched daily with fresh fruits, vegetables, grains, beans, nuts, and a multivitamin preparation. Semi-annual veterinary physical examinations were performed on all animals. Animals were socially housed and participated in an environmental enrichment program designed to encourage sensory engagement, enhance foraging behavior, novelty seeking, promote mental stimulation, increase exploration, play and activity levels, and strengthen social behaviors, together providing opportunities for animals to increase time budget spent on species typical behaviors. Monkeys were trained to cooperate in medical procedures including hand feeding and drinking, shifting into transport cages, and presentation for exam, drug administration, metabolic testing, and blood collection and instrumented with indwelling central and intraportal vascular access. Diabetes was induced with STZ (100 mg kg$^{-1}$ IV) and was confirmed by basal C-peptide <0.3 ng mL$^{-1}$ and negative C-peptide responses to IV glucose challenge. Monitoring of recipient monkeys included daily clinical assessments by study staff, regular evaluations by veterinary staff, and weekly hematology and chemistry laboratory studies. The care and treatment of all animals in this study were conducted with the approval of the University of Minnesota Institutional Animal Care and Use Committee and in compliance with the recommendations in the Guide for the Care and Use of Laboratory Animals (Institute of Laboratory Animal Resources, National Research Council, U.S. Department of Health and Human Services).

**ADL processing, release testing, and infusion**. On day −7 relative to islet transplantation, splenocytes were isolated from donor monkey spleens, red blood cells lysed, and remaining cells enriched for B cells with nylon wool columns (Polysciences, Inc.). The cells (80%) were agitated on ice for 1 h with ECDI (30 mg mL$^{-1}$ per $3.2 \times 10^8$ cells, AppliChem) in Dulbecco's phosphate-buffered saline (PBS), washed, cleaned of necrotic cells and microaggregates, and assessed for viability/necrosis by acridine orange/propidium iodide (PI) fluorescent microscopy. ECDI-fixed splenocytes were loaded into cold syringes ($n = 9$) or IV bags ($n = 2$) for IV infusion at a target dose of $0.25 \times 10^9$ cells per kilogram recipient BW with a maximum concentration of $20 \times 10^6$ cells mL$^{-1}$ and remained on ice until recipient administration. Induction of apoptosis was monitored in vitro by incubating ECDI-fixed cells at 37 °C for 4–6 hours, labelling with Annexin V/PI (Invitrogen), and analyzing by fluorescent microscopy.

To meet the target dose of ECDI-fixed ADLs for day +1 infusion, blood drawn from donor monkeys on days −15 and −7 relative to islet transplant and the remaining 20% of splenic cells were enriched for B cells via magnetic sorting using NHP CD20 beads (Miltenyi Biotec) and expanded ex vivo in a GREX100M flask (Wilson Wolf) until day +1 in the presence of rhIL-10 (10 ng mL$^{-1}$), rIL-4 (10 ng mL$^{-1}$), rhBAFF (30 ng mL$^{-1}$), rhTLR9a (10 ng mL$^{-1}$), and either rhCD40L-MEGA or both rhCD40L multimeric (500 ng mL$^{-1}$) and rhAPRIL (50 ng).

Expanded cells were stimulated with rhIL-21 (5 ng mL$^{-1}$), 24 h prior to harvest. The characteristics of ADL products infused into monkeys are presented in Supplementary Tables 15–18 (Cohort A, Cohorts C–E). Recipients were pretreated prior to infusion with a combination of diphenhydramine 12.5 mg, acetaminophen 160 mg, and ondansetron 4 mg per os (PO).

**Transient immunosuppression**. Immunosuppression was administered to all recipient monkeys in Cohorts A–E. To cover all ADL infusions in Cohort A, C, D, and E monkeys, first dose of each drug was given to all recipients in Cohorts A–E on day −8 or −7 relative to islet transplant on day 0. Antagonistic anti-CD40 mAb 2C10R4, provided by the NIH Nonhuman Primate Reagent Resource, was given IV at 50 mg kg$^{-1}$ on days −8, −1, 7, and 14. Rapamycin (Rapamune®) was given PO from day −7 through day 21 posttransplant; the target trough level was 5–12 ng mL$^{-1}$. Concomitant anti-inflammatory therapy consisted of: (i) αIL-6R (tocilizumab, Actemra®) at 10 mg kg$^{-1}$ IV on days −7, 0, 7, 14, and 21, and (ii) sTNFR (etanercept, Enbrel®) at 1 mg kg$^{-1}$ IV on days −7 and 0 and 0.5 mg kg$^{-1}$ subcutaneous on days 3, 7, 10, 14, and 21. Exploratory cohort monkeys were terminated at day +7, accordingly the last dose of immunosuppression was given to these monkeys on day +7.

**Islet processing, transplantation, and function**. Donor monkeys in Cohorts C–E underwent total pancreatectomy, and islets were isolated, purified, cultured for 7 days to minimize direct pathway stimulatory capacity, and subjected to quality control. On day 0, a target number of ≥5000 IE kg$^{-1}$ by DNA with endotoxin contents of ≤1.0 EU kg$^{-1}$ recipient BW (Supplementary Tables 19–22) was transplanted non-surgically using the indwelling intraportal vascular access port into STZ diabetic monkeys. Protective exogenous insulin was stopped at day 21 posttransplant in animals with full graft function. Metabolic monitoring included daily a.m./p.m. blood glucose, weekly C-peptide, monthly HbA1c, mixed meal testing, and bi-monthly IV glucose tolerance tests with determination of acute C-peptide response to glucose and glucose disappearance rate (Kg).

**Histopathology of islet grafts**. Liver specimens were obtained from 10 different anatomical areas in each recipient, fixed in 10% formalin, and processed for routine histology. Sections from each of the ten blocks were stained with hematoxylin & eosin or immunostained for insulin to score transplanted islets as described in Supplementary Tables 4–6. Rejection-free islet allograft survival was confirmed by demonstrating at necropsy on graft histopathology a considerable number of intact A-type and mildly infiltrated B-type islets with no or very few C- to F-type islets (moderately to markedly infiltrated islets and islets partially or completely replaced by infiltrates or fibrosis).

**Flow cytometric analysis of immune cell phenotypes**. Multicolor flow cytometric analyses were performed on cryopreserved PBL, LMNC, and LN samples of Cohort B–E monkeys. In all, $1 \times 10^6$ cells were stained with viability dye (Aqua; Life Technologies) to discriminate viable cells from cell debris. The cells were stained for 25 min at room temperature (RT) with antibodies, fluorescence-minus-one, and/or isotype controls, followed by fixation (eBioscience) and wash. To assess regulatory T cells and proliferating T and B cells and intracellular cytokines, PBLs were stained with antibodies recognizing extracellular epitopes (CD3, CD4, CD8, CD25, and CD127), followed by fixation/permeabilization with the FoxP3 Fixation/Permeabilization Kit (eBioscience) and staining with anti-FoxP3, Ki67, IFN-γ, IL-10, and TGF-β antibodies. A minimum of 200,000 events were acquired on three-laser BD Canto II (BD Bioscience) with FACSDIVA 6.1.3. Relative percentages of each of these subpopulations were determined using the FlowJo 10.1. software (TreeStar). Detailed gating strategies, antibody clones, and sources of antibodies are presented in Supplementary Figs. 4–7 and Supplementary Table 23.

**Gating strategy**. First, cells were gated on FSC-H vs. FSC-A and then on SSC-H vs. SSC-A to discriminate doublets. Lymphocytes were then gated based on well-characterized SSC-A and FSC-A characteristics. Dead cell were excluded based on viability dye. The following phenotypic characteristics were used to define immune cell populations: T cells: CD3$^+$ lymphocytes; CD4$^+$ T cells: CD4$^+$/CD3$^+$/CD8$^-$; CD8$^+$ T cells: CD8$^+$/CD3$^+$/CD4$^-$; CD4 or CD8 TEM cells were determined as CD27$^{hi}$/CD28$^-$ within CD4 or CD8 T cells. Expression of PD-1, Tbet, CD40 and Ki67 were determined on both CD4$^+$, CD8$^+$ T cells and CD20$^+$ B cells.

Chemokines receptor (CXCR-5) expression was examined on CD4 T cells to enumerate Tfh cells: CXCR5$^+$CD4$^+$ T cells. Regulatory T cells were defined as Tr1 cells: CD49b$^+$LAG-3$^+$ of gated CD4$^+$CD45RA$^-$ lymphocytes, Treg cells: CD127$^-$FoxP3$^+$ of gated CD4$^+$CD25$^+$ lymphocytes, NS cells: CD8$^+$CD122$^+$ of gated CD8 lymphocytes and for Breg cells: regulatory B cells (CD24$^{hi}$CD38$^{hi}$), B10 cells: (CD24$^{hi}$CD27$^+$) within CD3$^-$CD19$^+$/CD20$^+$ lymphocytes based on the expression of CD24, CD27, and CD38 antigens. Gated Lin$^-$(CD3$^-$CD20$^-$) HLA-DR$^-$ CD14$^+$ cells were analyzed to enumerate MDSC: CD11b$^{hi}$CD33$^{hi}$ of CD14$^+$ Lin$^-$HLA-DR$^-$ cells.

**Donor-specific T and B cell responses**. MLRs were performed on cryo-banked PBL samples from islet donors and transplant recipients. Responder PBLs (300,000 cells) samples from recipient monkeys were labeled with 2.5 μM CFSE (Invitrogen, Cat# C34554) and were cocultured with irradiated (3000 cGy) VPD-450-labeled (BD, Cat# 562158) stimulator PBLs (300,000 cells) from islet donors (donor) and unrelated MHC-mismatched donors (third-party).

In another set of experiment, CFSE-labeled PBL from naive responder monkeys were cocultured (300,000 cells) with ECDI-fixed PBLs (ADLs) from islet donors. ADLs were prepared as described in the ADL vaccine method. On day 6 of MLR, CFSE dilution was measured on CD4$^+$, CD8$^+$, and CD20$^+$ cells and presented in percentage of CFSE low cells as proliferative cells.

*Assessment of multifunctional cytokine profile of donor stimulated Tr1 cells.* T cells from the peripheral blood from Cohort B ($n = 3$), Cohort C ($n = 3$), and Cohort E ($n = 2$) monkeys were collected at the time of termination. Briefly, responder recipient PBLs ($1 \times 10^6$ cells) were cultured in the presence of donor PBLs (VPD-450 labeled) at 1:1 ratio for 48 h. These donor-primed cells were briefly activated with low dose of phorbol myristate acetate/Ionomycin for 4 h in the presence of Brefeldin-A (10 μg mL$^{-1}$). Cells were surface-stained for CD4, CD49b, and LAG-3 followed by permeabilization fixation and intracellular staining for IL-10 and TGF-β. Gating strategy to identify Tr1 cells was performed as described earlier.

*ELISPOT.* For IFN-γ ELISPOT assays, longitudinally collected PBLs from Cohorts B and C monkeys were thawed, washed, and preincubated in a 12-well culture plate at 37 °C, 5% CO$_2$ with donor PBLs in a final volume of 1 mL complete RPMI medium. After 48 h, cells were harvested, washed twice with PBS, and resuspended in 200 μL of culture medium. Cells were transferred to 2 ELISPOT wells coated with anti-IFN-γ antibody, and incubated in a final volume of 100 μL per well for 5 h at 37 °C. Subsequently, the ELISPOT assay (U-Cytech Biosciences) was executed according to the manufacturer's protocol. Spot analysis was performed with an Immune Spot ELISPOT reader (CTL, Cleveland, OH).

*IL-17 enzyme-linked immunosorbent assay (ELISA).* IL-17 determination in MLR culture supernatants was performed with the monkey IL-17A ELISA Development Kit (Mabtech, Cincinnati, OH) as per the manufacturer's instruction using a Synergy™ 2 Multi-Mode Microplate Reader (Biotek, VT, USA); IL-17 concentrations were extrapolated from a standard curve.

*Sensitization screening*. Sera from recipient monkeys were collected at different time points and the presence of DSAs was detected by flow cytometry. In brief, preserved donor PBLs were thawed and after washing with complete RPMI, re-suspended in $4 \times 10^6$ cells mL$^{-1}$ in fluorescence-activated cell sorting (FACS) buffer (PBS containing 2% fetal bovine serum). Fifty microliters of prepared cell suspension was seeded in each well of U-shaped 96-well plate along with 50 µL of complement deactivated (56 °C for 45 min) recipient's serum followed by 30-min incubation at RT and 3 times PBS wash, finally re-suspended in 100 µL of FACS buffer with FITC-anti-IgG, PE-anti-CD20, PE Cy7-anti-CD3, and LIVE/DEAD™ Fixable Aqua dye followed by incubation for 20 min at RT. After incubation, cells were washed twice, fixed with paraformaldehyde, and analyzed by BD FACS Canto II Flow Cytometer. Detection of anti-IgG levels on CD3$^+$ gated cells represent the amount of DSAs in each recipient's serum.

**Suppression assays examining immune regulation**. All designated regulatory subpopulations were sorted from Cohort C monkeys. PBLs obtained from freshly collected blood were labeled with CD4, CD49b, and LAG-3 for Tr1 cell (LAG-3$^+$ CD49b$^+$ of CD4$^+$) sorting, CD19, CD24, and CD38 for Breg cell (CD24$^+$CD38$^+$ of CD19$^+$) sorting, and CD4, CD25, and CD127 for Treg cell (CD25hi$^+$CD127$^-$ of CD4$^+$) sorting in sterile PBS followed by wash. The BD FACSAria II system was set up for a sort using an 85-µm nozzle (45 psi with a frequency of 47 kHz). All the sortings were performed at 8000–10,000 events per second. Sorted cells were collected in $12 \times 75$-mm round bottom tubes with CRPMI. Post sort analyses were performed for purity assessment.

In depletion assays, Treg, Breg, and Tr1 cells were depleted from PBLs of Cohort C monkeys, collected at 12 months posttransplant. Identical numbers of CFSE-labeled total PBLs (nondepleted) or Treg-depleted (non-CD4$^+$lym plus CD127$^-$CD25$^{hi}$CD4$^+$lym), Breg-depleted (non-CD19$^+$lym plus CD24$^-$CD38$^-$CD19$^+$lym), and Tr1-depleted (non CD4$^+$lym plus CD49b$^-$LAG3$^-$ CD4$^+$lym) PBLs were cultured with equal numbers of irradiated, VPD450-labeled donor PBLs in a 1-way CFSE Flow-MLRs for 6 days. For all CFSE-MLR proliferation assays, a 1:1 ratio of responder and stimulator cells was maintained. During flow analysis of proliferating cells (CFSE$^-$), the entire donor population was excluded based on VPD450$^+$ positivity.

To ascertain the suppressive capacity of sorted cells, naive recipient PBLs, collected at baseline before vaccination and transplantation, were challenged with irradiated VPD450-labeled donor PBL cells (1:1 ratio) in 1-way CFSE Flow-MLR for 3–4 days followed by re-challenge with irradiated, donor PBLs in the presence or absence of various types and ratios (1:50) of immune cells with regulatory phenotypes (Tr1, Treg, and Breg cells). These cells were sorted from tolerant recipients between 9 and 12 months posttransplant. For all suppression assays examining Tr1 cells, we used at 1:50 ratio of Tr1 vs. total PBLs in the presence of donor and third-party donor.

Transwell experiments were set up to study whether Tr1-mediated suppression is contact dependent. In this set of experiments, CFSE-labeled Tr1-depleted PBLs were seeded (300,000 cells) in the bottom of the plate with irradiated, VPD450-labeled donor cells (1:1 ratio) in the presence or absence or Tr1 cells separated by the transmembrane (4-µm pore size, Corning, Ref #3391), either in the presence (10 µg mL$^{-1}$) or absence of anti-human IL-10 neutralizing Ab, known to crossreact with IL-10 of monkeys, and matched isotype.

**siRNA-mediated SH2D2 inhibition in Tr1 cells**. Flow sorted Tr1$^+$ cells (CD49b$^+$ LAG-3$^+$CD4$^+$) and Tr1$^-$ lym cells (pool of CD4$^-$lym + CD49b$^-$ LAG-3$^-$ of CD4$^+$) were sorted from PBLs of Cohort C monkeys collected at 12 months posttransplant. CFSE-labeled Tr1$^-$ lym cells (300,000) were cultured with or without VPD-450-labeled irradiated donor cells (300,000) for 6 days MLR. Initially, sorted Tr1 cells are rested for first 3 days in CRPMI and later they were treated with 100 µM Accell Human SH2D2A siRNA (Dharmacon Accell, Cat# E-017851-00-0005) by combining Accell siRNA stock solution and Accell delivery media (GE Healthcare, Cat# B-005000-500) directly to sorted Tr1$^+$ cells.

Tr1$^+$ cells treated with SH2D2A siRNA or Accell delivery media alone were added back to MLR for last 3 days. To measure the impact of siRNA-mediated SH2D2 inhibition on Tr1-mediated suppression of donor-specific T and B cells, on day 6 total culture cells were harvested and stained for assessment of T and B cell proliferation.

**Tetramer preparation and staining**. To enable tracking of CD4$^+$ T cells, Tr1 cells, and Treg cells with indirect allopeptide specificity in these monkeys with MHC class II tetramers, we exploited the high degree of similarity observed in the peptide-binding motifs of MHC class II molecules in rhesus monkeys, cynomolgus monkeys, and humans. We performed a t-BLAST analysis of the *Mamu DRB* sequence with the human genome at the NCBI website to determine the human homolog. HLA DRB1*13 (Acc. No. CDP32905.1) was 92% identical, with 96% positives and 0% gaps to the *Mamu DRB03a* with an *e* value of 6e-178 and HLA DRB1*14 (Acc. No. ABN54683.1) was 93% identical, with 94% positives and 0% gaps to the *Mamu DRB04* with an *e* value of 2e-175.

Peptides from *Mamu* MHC class I and class II sequences with high binding affinity for HLA DRB1*13 or HLA DRB1*14 were identified (Supplementary Table 24) using the Immune Epitope Database Analysis resource. Synthetic

peptides (Genscript USA Inc.) were loaded onto the HLADRB1*13 or HLA DRB1*14 tetramers. PBL were incubated with 0.5 or 1 µg mL$^{-1}$ HLA class II tetramer PE along with the antibodies for specific cell surface markers for 20 min. Stained cells were washed with cold PBS supplemented with1% fetal calf serum, fixed in 2% formaldehyde, acquired on BD Canto II, and data analysis was performed using FlowJo version 10.2 (Tree Star, Ashland, OR).

**TCR sequencing for tracking donor-reactive T cells**. We employed a RNA-based, high-throughput sequencing of the TCR β chain CDR3 region to compare the entire repertoire of T cell clones at intervals before and after ADL infusions in Cohort A monkeys. This approach has the advantage over genomic methods that require designing and optimizing multiplex primer sets that span the entire V gene segment; these remain poorly defined in monkeys. Total RNA was extracted from frozen PBLs using RNeasy Plus Universal (Qiagen) and first-strand cDNA was created from the poly A tailed fraction of total RNA. Briefly, custom-designed oligo dT primer and a template switching primer were used in a reverse transcriptase reaction to synthesize cDNA, which was used as template for targeted PCR enrichment of the TCR VDJ region using primers specific to the *M. mulatta* TCR constant region and the template switch sequence (Supplementary Table 25). Enriched VDJ amplicons from each sample were uniquely dual-indexed by PCR for multiplexing compatibility during sequencing. All PCR amplifications were performed with the KAPA HiFi HotStart DNA Polymerase ReadyMix Kit (Roche). Indexed amplicon libraries were pooled equimolarly and cleaned with SPRI beads (Ampure XP, Beckman Coulter). The final pool was sequenced on a MiSeq (Illumina) 300 bp paired end run (v3 kit). Preparation and sequencing of TCR NGS libraries was performed at the University of Minnesota Genomics Center. Raw, QC-ed short reads were cleaned via trimmomatic using set parameters (ILLU-MINACLIP:all_illumina_adapters.fa:2:30:10 LEADING:3 TRAILING:3 SLI-DINGWINDOW:4:15 MINLEN:70). The preprocessed reads were directly input into MiXCR for TCR profiling with default setting (https://mixcr.readthedocs.io/en/latest/rnaseq.html). Known TRB clonotypes in rhesus monkeys was used as reference. The resulted TRB clonotypes were further filtered using customized threshold with a clone fraction of ≥0.5%. A barcode was included during the cDNA synthesis process that uniquely tags each RNA molecule that was reverse transcribed. The template switching oligo Hrg004_TempSwUID used during cDNA synthesis has a barcode sequence DDDDDTGTDDDDDTGTDDDDD where D = A or G or T (barcode diversity is $3^{15}$ or ~14.3 million sequences). The barcode is part of the Read 1 sequence data. The TCR sequencing analysis software used corrects for PCR amplification bias using the barcode sequence, i.e., all sequences that share the same barcode are considered as a single sequence for analysis. Frequency of clonal expansion was calculated by dividing the frequency of the clone at individual time points over the average frequency of all the identified mapped TCR clones. Most of the donor-specific TCR clones at the baseline were very low or undetectable.

**RNASeq for examining gene expression in sorted Tr1 cells**. RNA samples were sequenced using the Illumina Hiseq 2500 platform 50 bp paired end reads. Raw sequence that passed CASAV 1.8P/F filter were assessed by fastqc (http://www.bioinformatics.babraham.ac.uk/projects/fastqc). Read mapping was performed via Hisat2 (v2.0.2) using the UCSC human genome (hg38) as reference. Cuffdiff 2.2.1 was used to quantify the expression level of each known gene in units of FPKM (fragment mapped per kilobase of exons per million mapped reads). Differentially expressed genes were identified using the edgeR (negative binomial) feature in CLCGWB (Qiagen, Valencia, CA) using raw read counts. DEG is presented on a color scale. The expressed transcripts were annotated using the NHP or Human Genome database. The results were ranked by the absolute value of fold change, and DEG between Cohort B and Cohort C were identified. We filtered the generated list based on a minimum 1.5× Absolute Fold Change and raw *p* values <0.01. These DEGs were imported into the Ingenuity Pathway Analysis Software (Qiagen, Valencia, CA) for pathway identification. Hypergeometric distribution was utilized to determine whether certain pathways were overrepresented (enriched) in the DEGs using the Reactome knowledge base corrected for false discovery rate using the Benjamani–Hochberg method.

**Statistical analysis**. Statistical analysis of flow cytometric data distributions of values within all groups were checked for Gaussian distribution. Owing to the small sample size and non-normal distribution, we used the non-parametric Mann–Whitney *U* test followed by post hoc analysis with the Holm–Sidak method for comparisons between two groups and unpaired *t* test with Welch's correction. Statistical analysis using paired *t* test was used to compare within groups (naive vs. treatment). Time to event (rejection) curves for each group were estimated by using the Kaplan–Meier method and compared using log-rank (Mantel–Cox) test. Groups were considered as significantly different when $P < 0.05$. Statistical analysis of transcriptomic data of gene differential expression were performed using an empirical Bayes moderated *t* statistic, with a cutoff of 0.05, corrected for multiple hypotheses testing using Benjamini–Hochberg procedure, and an absolute fold change cutoff of >1.4 with the limma package. Analyses were performed using R (R Foundation for Statistical Computing; Vienna, Austria; Version 3.4.2.). Graphs were generated with GraphPad Prism 7.

**Reporting summary**. Further information on research design is available in the Nature Research Reporting Summary linked to this article.

## Data availability

The RNA-seq data of the Tr1 cells (accession No. GSE132691) and the TCR sequencing data (accession No. GSE132496) are available at NCBI in GEO.

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

## Acknowledgements

We gratefully acknowledge the excellent and expert care, husbandry, and training of our animals by the team at the University of Minnesota's Preclinical Research Center, coordinated by Lucas Mutch and Jody Janecek. We thank Brian E. Flanagan, Jean Witson, Kate Mueller, Thomas Gilmore, Anders Matson, Zach Swanson, and Steven Kass of the Schulze Diabetes Institute at the University of Minnesota; Roger Wiseman of the Wisconsin National Primate Research Center's Genetics Services Unit; and I-Ting Chow and Cynthia Cousens-Jacobs of the Tetramer Core Facility at the Benaroya Research Institute at Virginia Mason. We thank Keith A. Reimann of the NIH Nonhuman Primate Reagent Resource (R24 OD010976, U24 AI126683) for providing the anti-CD40 antibody 2C10R4; Timothy D. O'Brien of the Comparative Pathology Section in the University of Minnesota's Department of Veterinary Population Medicine, faculty/staff members of the University of Minnesota's Genomics Center for consultation and for execution of TCR sequencing and transcriptome studies; faculty/staff members of Northwestern University's HLA laboratory, directed by Anat Tambur, for their assistance in measuring PRA and DSA; and Mary Knatterud, PhD, of the University of Minnesota's Department of Surgery, and Neeta Adhikari, PhD, for editing the manuscript. Research reported in this publication was supported by the National Institute of Allergy and Infectious Diseases of the National Institutes of Health as part of the Nonhuman Primate Transplantation Tolerance Cooperative Study Group under Award Number U01AI102463. The content is solely the responsibility of the authors and does not necessarily represent the official views of the National Institutes of Health. This work was also supported by funds provided by the Diabetes Research and Wellness Foundation, the Transplant Division in the University of Minnesota's Department of Surgery, and individual philanthropy through the University of Minnesota Foundation.

## Author contributions

A.S. designed immune monitoring assays, performed immune mechanistic studies, manufactured and analyzed ADL products, analyzed and interpreted data, and wrote the manuscript. S.R. designed immune monitoring assays and TCR sequencing, established the tetramer technology, performed immune mechanistic studies, analyzed and interpreted data, and wrote the manuscript. M.L.G. conducted and directed the studies in monkeys, performed all surgeries and necropsies on donors and recipients, performed the metabolic studies, analyzed and interpreted data, and wrote the manuscript. S.D. performed immune mechanistic studies, analyzed and interpreted data, and edited the manuscript. D.H. and W.L.S.-P. manufactured and analyzed ADL products. A.N.B., J.D.A., and J.J.W. manufactured and analyzed islet products. A.Y. performed statistical analyses. Y.Z. developed computational methods and analyzed the TCR sequencing data. N.P.P. and S.R. designed and conducted TCR sequencing. J.E.A. performed transcriptomic analyses. C.B. designed assays, interpreted data, and edited the manuscript. S.D.M. and X.L. conceived and co-supervised the study, interpreted the data, and edited the manuscript. B.J.H. conceived and supervised the study and coordinated its conduct, developed methods for manufacturing of ADL products, designed the immunosuppressive regimen, planned the immune mechanistic assays, interpreted the data, and wrote the manuscript with input from all the authors.

## Additional information

**Competing interests:** S.R. and C.B. received research support for projects not reported in this article by Diabetes-Free, Inc., an organization that may gain or lose financially through this publication. M.L.G. is a paid consultant for Otsuka Pharmaceutical Factory, Inc. S.D.M. is a co-founder, member of the SAB and a paid consultant for Cour Pharmaceuticals Development Company; a member of the SAB and a paid consultant for NextCure Inc.; and a paid consultant for Kite Pharmaceuticals. S.D.M. and X.L. are inventors on issued patent no. US 8,734,786 B2 submitted by Northwestern University that covers the use of ECDI-fixed cell tolerance as a method for preventing allograft rejection. B.J.H. has an equity interest in and serves as an executive officer of Diabetes-Free, an organization that may commercially benefit from the results of this research. This interest has been reviewed and managed by the University of Minnesota in accordance with its Conflict of Interest policies. The other authors declare no competing interests.

