## [Peer Review File · Nature Communications]

Reviewers' comments:

Reviewer #1 (Remarks to the Author):

The authors showed that two peritransplant infusions of apoptotic donor leukocytes under short-term immunotherapy with antagonistic anti-CD40 antibody 2C10R4, rapamycin, soluble tumor necrosis factor receptor, and anti-interleukin 6 receptor antibody induced long-term (≥ 1 year) tolerance to islet allografts in 5 of 5 nonsensitized, MHC class I-disparate, and 1 MHC class II DRB allele-matched rhesus macaques. This regimen induced the potent and sustained immunoregulation in-vivo by inducing several regulatory cell types and depleting donor-specific T, B cells. The findings are quite interesting and the experiments are well-designed. However, some areas need additional information.

1) Fig 1. Authors concluded that ALD infusion induces abortive expansion of donor specific T and B cells. However, authors only analyzed PBLs from cohort A group. To clarify whether these effects solely depend on ALD infusion, not other immunosuppressants, analysis of control group (w/o ALD infusion, with Immunosuppressants) should be included.

2) The authors showed that Tr1 cells were expanded in cohort C group. Although this is critical point of this study, they did not show the underlying mechanisms of this expansion. APCs might be crucially involved in the induction or expansion of Tr1.

- Are there any changes in APCs?

- APCs from cohort C express higher level of PDL-1/2 than those from cohort B? or produce specific cytokines?

3) The authors mention that there was no significant difference between cohort B and C in frequency of IFN- γ secreting T cells in response to irradiated donor PBLs. They have no discussion of this point and the manuscript would benefit from such a discussion.

4) In Fig 4, and Supplementary Fig 12, authors showed the differentially expressed genes in Tr1 cells from Cohort C. Adding the Enrichment map would be more informative.

Reviewer #2 (Remarks to the Author):

Long-term tolerance of islet allografts in nonhuman primates induced by apoptotic donor leukocytes, by Amar Singh, et al.

This manuscript describes the induction of long-term islet function in rhesus macaques using a short-course of induction combined with the infusion of ethylcarbodiimide-treated apoptotic donor leukocytes. The overall quality of the manuscript is very good and the report describes a highly-significant finding in a challenging rhesus macaque model. Given the overall excellent quality of the manuscript, I have several comments and criticisms that are detailed below.

1. The data presented in Figure 1 would be more significant if there was a control group that received the immunosuppression induction regimen without the infusion of apoptotic donor leukocytes since several of the agents used might be able to modify the activation and division of many of the cell types analyzed.

2. In the main text, in reference to Figure 1d, the authors state that the frequency of interferon-gamma-secreting CD4 T cells dropped significantly. However, while the drop is impressive in the figure,

- they should show a statistical analysis of the change in order to say the drop was significant.
3. Likewise, for the discussion of donor-specific proliferation in Fig 1e-g, statistics need to be run/shown to claim the proliferation dropped significantly.
 4. The clone frequency section is an interesting analysis, but has a few pitfalls/assumptions that are not explicitly acknowledged. When generating the cDNA library from the RNA, was a bar code added to the cDNA? This is a technique to control for PCR amplification bias during PCR enrichment. If not, certain clones may have been more enriched than others during the enrichment step leading to a false interpretation of clonal expansion. Also, there may be more copies of the RNA in activated cells that introduces further bias in this analysis. In my experience, it is also difficult to capture all the clones in this method of going from RNA->cDNA as we often obtain fewer numbers of clones from unstimulated cells than from the same cells after 5-days of stimulation with donor cells. Also, there is an assumption that the alteration in clonal frequency is due entirely to ADL infusions. Was this analysis done in Group B that didn't receive ADLs, thus lending this assumption more weight if no clones in that cohort had significant changes? I think the analysis is acceptable as long as some of these limitations are noted in the discussion. Since there are several other analyses that point toward a donor-specific response, the limitations of this analysis do not alter the conclusions of the paper.
 5. The main text referencing Figure 1J states that T-cell clone analysis demonstrated abortive expansion and subsequent contraction. However, since the baseline is not shown (only days -5 and -4) the figure only shows the contraction and not abortive expansion. Showing the analysis preceding D-5 would remedy this.
 6. In Figure 3, the %TEM in Group B and Group C (Fig 3a and 3d) differed in the 3- and 6-month analysis, but approximated each other by 12 months. However, the analysis of the liver mononuclear cells and lymph nodes was performed at the time of rejection, which varied for Group B, with several animals being sacrificed before 1 year. Is it possible that the differences in 3b and 3e could be due in part to the different time point at which the analyses were performed?
 7. At the bottom of page 8, it reads, "...were 2 major biological pathways activated in sorted Tr1 cells in Cohort C, but Cohort B RM (Supplementary Table 7)". There seems to be a word missing in the segment I underlined.
 8. In the segment on Sensitization interferes with the tolerogenic efficacy of ADLs, there is discussion on how ADLs cultured with PBLs from sensitized animals stimulate greater proliferation of donor-specific CD4 and CD8 T cells. Previously, the authors stated that the stimulation resulted in abortive expansion with in vivo data showing PD1+ cells that may be anergic. Was any analysis performed on the donor-specific cells in sensitized animals to see if they were behaving differently and were not exhausted/anergic but activated? Otherwise it is hard to draw conclusions from this section.
 9. Supplementary Figure 3: The legend does not (solid red squares and solid blue circles) is inconsistent with the figure (solid blue squares and open red circles).

Reviewer #3 (Remarks to the Author):

The manuscript by Singh and colleagues demonstrates the feasibility of inducing donor-specific tolerance to MHC class I mismatched, 1 DRB-class II-matched islet allografts in non-human primates with STZ-induced diabetes, without having to co-transplant bone marrow or induce donor chimerism. The authors use infusions of apoptotic donor leukocytes (ADL), in a regimen that could have easier clinical applicability than the regimens trying to induced mixed chimerism currently in clinical trials. However, the authors' regimen is less effective in full mismatched islet transplantation or in sensitized hosts. Mechanistically, they argue that ADL induces a regulatory network that suppresses residual donor-specific effector T cells.

The long-term allograft acceptance in the class I mismatch, 1 DRB-matched islets is quite impressive,

with evidence of true donor-specific tolerance which is remarkable; the analysis of donor-specific responses using pMHC tetramers and functional readouts in vitro is elegant; and the characterization of the Tr1 cells and siRNA manipulation is convincing to infer the induction of regulatory mechanism. However, some issues limit enthusiasm for the current manuscript.

1. The authors ascribe their results to the use of apoptotic donor leukocytes (ADL) and they have extensive experience with the tolerogenic potential of ECDI-fixed cells, from mouse models of autoimmunity and transplantation to clinical trials in multiple sclerosis. However, in this particular study, the monkeys in the control group (cohort B) received transplantation without immunosuppression, whereas the experimental group (cohort C) received ADL 2x plus blocking anti-CD40 plus rapamycin plus sTNFR and plus anti-IL-6R. The fact that 2/5 monkeys in the untreated control group accepted their allografts long-term implies that the immunogenicity of this donor/recipient combination is modest. Therefore, it is not possible to determine if the effects observed with the immunosuppressive treatment are in fact due to the ADL infusion, or to the other components of the immunosuppressive regimen, or to the combination of both, unless the authors have another control cohort that receives everything except ADL (and perhaps a control cohort that receives ADL alone, though this one is less crucial). This reviewer appreciates the cost of additional cohorts, but the authors cannot make the claims they make without them.

2. As a continuation of the previous point, if the full immunosuppressive regimen had been more effective in cohorts D (full mismatch) and E (sensitized), it may not have mattered as much to tease out the role of ADL versus the other drugs, as one might imagine moving the whole combo forward to the clinic. But because donor-specific tolerance is only achieved in the weaker combination, such that the regimen will need to be further optimized for more immunogenic donor/recipient combinations, it is important to understand the additional impact of ADL over that of the other biologics.

3. The authors seem to overinterpret some of their findings. In particular, they claim that ADL induces Bregs. To the knowledge of this reviewer, there still isn't a marker that faithfully identifies Bregs in the way FoxP3 can identify Tregs. The authors use CD24+CD38+ as their definition of Bregs, but this likely only identifies activated B cells. It is possible that they contain a subset of Bregs, but it seems overinterpreted to call these Bregs. Similarly for B10 cells and natural suppressor cells. Although it is possible, as the authors state, that ADL induces a regulatory network of Tregs, Tr1s, Bregs, MDCs etc, the evidence here, except for the Tr1 cells, is weak.

Reviewers' Comments:

Reviewer #1 (Remarks to the Author):

The authors showed that two peritransplant infusions of apoptotic donor leukocytes under short-term immunotherapy with antagonistic anti-CD40 antibody 2C10R4, rapamycin, soluble tumor necrosis factor receptor, and anti-interleukin 6 receptor antibody induced long-term (≥ 1 year) tolerance to islet allografts in 5 of 5 nonsensitized, MHC class I-disparate, and 1 MHC class II DRB allele-matched rhesus macaques. This regimen induced the potent and sustained immunoregulation in-vivo by inducing several regulatory cell types and depleting donor-specific T, B cells. The findings are quite interesting and the experiments are well-designed. However, some areas need additional information.

1) Fig 1. Authors concluded that ALD infusion induces abortive expansion of donor specific T and B cells. However, authors only analyzed PBLs from cohort A group. To clarify whether these effects solely depend on ALD infusion, not other immunosuppressants, analysis of control group (w/o ALD infusion, with Immunosuppressants) should be included.

Response:

We agree with the reviewer that analyzing PBLs from Cohort A alone will not clarify whether the abortive expansion of donor-specific T and B cells solely depends on ADL infusions and not on concomitantly administered immunosuppressants. A related question is whether abortive expansion is in part or entirely secondary to concomitant immunosuppression. To address these important questions in more detail, as suggested by Reviewers 1 and 2 (see below), we have considered analyzing an additional “mechanistic control Cohort A-2” that would only receive immunosuppression but no ADL infusions. As we thought more closely through the experimental design, it became apparent that to rule out that the abortive expansion of donor-specific T and B cells observed in Cohort A was secondary to the immunosuppressants used in our study would actually require the administration of donor antigen. Without providing donor antigen pretransplant or peritransplant on days -7 and + 1, it would seem impossible to ascertain whether donor-specific deletion of T and B cells is mediated in a control cohort solely by the immunosuppressants.

To address the reviewer’s question without this experimental limitation, we directly compared - at an early time point posttransplant - the clone-sizes of donor-specific T cells in Cohorts B and

C, i.e. in monkeys that received immunosuppression and donor islets without ADLs (Cohort B) and with ADLs (Cohort C). To determine donor-specific T cell expansion and depletion associated with the treatment protocols in Cohorts B and C on day 7 posttransplant, we used recipient-type MHC class II tetramers loaded with mismatched donor MHC class I peptides to compare the fold-changes (versus baseline) of the percentages of CD4+ T cells with indirect donor MHC class I specificity among the subset of circulating CD4+ T cells with non-regulatory phenotypes. As shown in the new **Supplementary Fig. 8I**, and referred to in the revised Results section (**page 8, lines 13-23**), that average fold-change in 3 Cohort B monkeys was 2.55 ± 0.09 , whereas there was no evidence of expansion of donor-specific, indirect CD4+ T cells in the 2 studied Cohort C monkeys (fold-change of 0.92 ± 0.014). These findings do not provide direct evidence that immunosuppression alone failed to facilitate abortive expansion of donor-specific T cells pretransplant. However, our new results show important differences between the 2 key Cohorts B and C and are consistent with the interpretation that the concomitant immunosuppression used in this study was by itself not mediating the abortive expansion of anti-donor T cells presented in Cohort A monkeys in Fig. 1.

2) The authors showed that Tr1 cells were expanded in cohort C group. Although this is critical point of this study, they did not show the underlying mechanisms of this expansion. APCs might be crucially involved in the induction or expansion of Tr1.

- Are there any changes in APCs?

- APCs from cohort C express higher level of PDL-1/2 than those from cohort B? or produce specific cytokines?

Response:

Reviewer #1 is correct in pointing out the importance of APCs in the generation of Tr1 cells.

In response to the reviewer's questions about changes in APCs, we analyzed additional data to determine the effects of ADL infusions on APC subsets. Interestingly, when comparing Cohorts B and C, ADL infusions were associated with downregulation of HLA-DR expression in CD11b+ DCs, CD14+ monocytes, but only marginally in CD20+ B cells at 2 and 4 weeks posttransplant, whereas HLA-DR expression increased in all three APC subsets in control Cohort B animals (new **Supplementary Fig 10. c-e**). Furthermore, we added additional data on the effect of ADL infusions on the percentage of circulating MDSCs on day 14 posttransplant, showing a

substantial increase in Cohort C (from $22.86 \pm 6.20\%$ to $47.74 \pm 15.48\%$ of $CD14^+Lin^-HLA-DR^-$ cells) and only a small increase in circulating MDSCs in Cohort B (from $17.65 \pm 5.80\%$ to $24.01 \pm 10.45\%$ of $CD14^+Lin^-HLA-DR^-$ cells, revised **Supplementary Fig 10b**). These findings extend the results on effects of ADL infusions on circulating MDSCs in Cohort A (**Fig. 1b**). (**Page 9, lines 7-16**)

Additional studies on APC subsets in Cohort A revealed a profound downregulation of circulating $HLA-DR^+$ monocytes from $87.73 \pm 4.68\%$ at baseline to $55.83 \pm 10.69\%$ at 3 days after the first ADL infusion (new **Supplementary Fig. 1a**). Additionally, ADL infusions and immunosuppression were associated with a reduction in the $HLA-DR$ MFI on circulating monocytes from 385.67 ± 59.28 at baseline to 227.66 ± 52.53 at 3 days after the first ADL infusion. Shortly after ADL infusions, immunosuppressed Cohort A monkeys also showed considerably lower percentages of $CD80^+$ monocytes and DCs (new **Supplementary Fig. 1b-c**) and increased percentages of $PD-L1^+$ monocytes and DCs (new **Supplementary Fig. 1d-e**). (**Page 5, lines 10-15**)

To address the critical involvement of APCs in Tr1 cell generation, we have added the following text to the Discussion:

“It is now well established that Tr1 cell generation is mediated by APCs. Tr1 cells are induced in the periphery from naïve $CD4^+$ T cells upon TCR stimulation by APCs under tolerogenic conditions in an IL-10-enriched microenvironment; with distinct subsets of APCs, e.g. DC-10 cells, being the major source of IL-10 (Gregori, Tomasoni et al. 2010, Roncarolo, Gregori et al. 2014, Roncarolo, Gregori et al. 2018). Previous studies demonstrated IL-10 production triggered by apoptotic debris (Chung, Liu et al. 2007) and rapid and sustained IL-10 release from splenic marginal zone APCs after their uptake of intravenously infused, ECDI-treated, apoptotic leukocytes (Getts, Turley et al. 2011). ADL infusions also altered the phenotype of APCs in our study, suggestive of their possible involvement in Tr1 cell induction and expansion.” (**Page 14, lines 29-34 and Page 15, lines 1-2**).

3) The authors mention that there was no significant difference between cohort B and C in frequency of $IFN-\gamma$ secreting T cells in response to irradiated donor PBLs. They have no discussion of this point and the manuscript would benefit from such a discussion.

Response:

To address this point, we have inserted the following text into the revised Discussion:

“Analysis of donor specific proliferation showed a significant difference in proliferation of proliferation of CD4⁺ and CD8⁺ T cells at 1 month and 3 months posttransplant (**Fig 3c,f**). However, we did not see a significant difference between Cohorts B and C in the frequency of IFN- γ -secreting T cells in response to irradiated donor PBLs in ELISPOT assays. Several studies have demonstrated that donor-specific alloimmune responses can be mediated through the production of cytokines other than IFN- γ including IL-6 (Jordan, Choi et al. 2017, Uehara, Solhjou et al. 2018) and IL-17 (Kwan, Chadban et al. 2015, Matignon, Aissat et al. 2015, Agashe, Jankowska-Gan et al. 2018) and that CD4⁺ T cells with indirect specificity can mediate skin graft rejection in the absence of IFN- γ (Valujskikh and Heeger 2000, Marino, Paster et al. 2016).” (**Page 14, lines 10-18**).

4) In Fig 4, and Supplementary Fig 12, authors showed the differentially expressed genes in Tr1 cells from Cohort C. Adding the Enrichment map would be more informative.

Response:

A statistical (hypergeometric distribution) test was used to determine whether certain Reactome pathways are over-represented (enriched) in the differentially expressed genes using the Reactome knowledge base developed by Fabregat et al. (Fabregat, Jupe et al. 2018). The produced probability scores were corrected for FDR using the Benjamini-Hochberg method. Based on the enrichment analysis, per the reviewer’s suggestion, the genes were grouped based on the pathways impacted and are now presented in the revised **Supplementary Table 6. (Page 9, lines 31-33 and Page 24 lines 34-35 and page 25, lines 1)**.

Reviewer #2 (Remarks to the Author):

Long-term tolerance of islet allografts in nonhuman primates induced by apoptotic donor leukocytes, by Amar Singh, et al.

This manuscript describes the induction of long-term islet function in rhesus macaques using a short-course of induction combined with the infusion of ethylcarbodiimide-treated apoptotic donor leukocytes. The overall quality of the manuscript is very good and the report describes a highly-significant finding in a challenging rhesus macaque model. Given the overall excellent

quality of the manuscript, I have several comments and criticisms that are detailed below.

1. The data presented in Figure 1 would be more significant if there was a control group that received the immunosuppression induction regimen without the infusion of apoptotic donor leukocytes since several of the agents used might be able to modify the activation and division of many of the cell types analyzed.

Response:

We agree with Reviewer #2. As mentioned in our response to a very similar suggestion raised in Point #1 by Reviewer #1, it would be very helpful to dissect the contributions made by immunosuppression alone. Because **Figure 1** focuses predominantly on donor-specific T and B cells, we think that monitoring these cells properly without the administration of antigen provided by donor cells is very challenging.

Therefore, we feel strongly that the best possible control group in which to study the effects of immunosuppression without ADLs is Cohort B. We have inserted new data on early posttransplant frequencies of donor-reactive CD4⁺ T cells with indirect specificity for mismatched donor MHC class I peptides in both Cohort B and C animals (**Supplementary Fig. 8I**, and referred to in the revised Results section). These data show a 2.5-fold expansion of donor-specific CD4⁺ T cells by day 7 posttransplant, thus strongly suggesting that immunosuppression alone without ADL infusions is ineffective in mediating abortive expansion of donor-specific T cells. In contrast, there was no expansion of that subset at day 7 in the Cohort C monkeys. (**Page 8, lines 13-23**).

2. In the main text, in reference to Figure 1d, the authors state that the frequency of interferon-gamma-secreting CD4 T cells dropped significantly. However, while the drop is impressive in the figure, they should show a statistical analysis of the change in order to say the drop was significant.

Response:

Statistical analysis using paired T test was used to analyze whether a significant reduction in IFN- γ secreting CD4⁺ T cells was observed after ADL infusions when compared to naïve animals. Statistically significant differences have been marked in the revised **Figure 1d** (P= 0.038 at d0, P= 0.038 at D+3, P= 0.040 at D+7). (**Page 25, lines 7-8**).

3. Likewise, for the discussion of donor-specific proliferation in Fig 1e-g, statistics need to be run/shown to claim the proliferation dropped significantly.

Response:

Statistical analysis using paired T test was used to analyze whether a significant reduction was observed after ADL infusions when compared to naïve animals. Statistically significant differences were as follows

CD4+ T cells P= 0.024 at d0, P= 0.012 at d+3, and P= 0.001 at d+7

CD8+ T cells P= 0.012 at d0, P=0.003 at d+3, and P= 0.031 at d+7

CD20+ B cells P= 0.034 at D-4, P=0.0004 at d0, 0.0047 at d+3, and P= 0.005 at d+7

and have been marked in the revised **Figure 1. (Page 25, lines 7-8).**

4. The clone frequency section is an interesting analysis, but has a few pitfalls/assumptions that are not explicitly acknowledged. When generating the cDNA library from the RNA, was a bar code added to the cDNA? This is a technique to control for PCR amplification bias during PCR enrichment. If not, certain clones may have been more enriched than others during the enrichment step leading to a false interpretation of clonal expansion. Also, there may be more copies of the RNA in activated cells that introduces further bias in this analysis. In my experience, it is also difficult to capture all the clones in this method of going from RNA->cDNA as we often obtain fewer numbers of clones from unstimulated cells than from the same cells after 5-days of stimulation with donor cells. Also, there is an assumption that the alteration in clonal frequency is due entirely to ADL infusions. Was this analysis done in Group B that didn't receive ADLs, thus lending this assumption more weight if no clones in that cohort had significant changes? I think the analysis is acceptable as long as some of these limitations are noted in the discussion. Since there are several other analyses that point toward a donor-specific response, the limitations of this analysis do not alter the conclusions of the paper.

Response:

We thank Reviewer #2 for the insightful suggestions. The additional information provided below has been included in the paragraph “TCR sequencing for tracking donor-reactive T cells” in the Methods section of the revised manuscript:

A barcode was included during the cDNA synthesis process that uniquely tags each RNA molecule that was reverse transcribed. The template switching oligo Hrg004_TempSwUID used during cDNA synthesis has a barcode sequence DDDDDTGTDDDDTGTDDDDD where D = A or G or T (barcode diversity is 3^{15} or ~ 14.3 million sequences). The barcode is part of the Read 1 sequence data. The TCR-sequencing analysis software used corrects for PCR amplification bias using the barcode sequence; i.e., all sequences that share the same barcode are considered as a single sequence for analysis (**Page 24, lines 9-16**).

Another potential limitation of this analysis is the assumption that increased RNA copy number directly correlates with increase in the frequency of clones. To address this point and other related points, we have added the following text to the Results section. Further in-depth sequencing, single cell RNAseq, and comparative analysis with Cohort B (only immunosuppressed) are warranted to confirm these interesting findings (**Page 6, lines 4-7**).

5. The main text referencing Figure 1J states that T-cell clone analysis demonstrated abortive expansion and subsequent contraction. However, since the baseline is not shown (only days -5 and -4) the figure only shows the contraction and not abortive expansion. Showing the analysis preceding D-5 would remedy this.

Response:

In the revised **Fig. 1j**, though we could only detect a few of the clones in the naïve animals, we have included TCR frequencies of the clones detected pre-ADL infusions to demonstrate the abortive expansion and subsequent contraction of these clones. (**Page 6, lines 3-4**).

6. In Figure 3, the %TEM in Group B and Group C (Fig 3a and 3d) differed in the 3- and 6-month analysis, but approximated each other by 12 months. However, the analysis of the liver mononuclear cells and lymph nodes was performed at the time of rejection, which varied for Group B, with several animals being sacrificed before 1 year. Is it possible that the differences in 3b and 3e could be due in part to the different time point at which the analyses were performed?

Response:

We agree with the reviewer that the differences in the timing of sacrifice of recipients and corresponding sample collection could have contributed to the differences presented between Cohorts B and C in Fig. 3b and 3e. While we have no direct data to support this point, we think it is fair to assume that, if the percentages of CD4+ and CD8+ TEM cells within LMNCs and LNs were low at 1 year or later posttransplant in tolerant Cohort C monkeys as shown in **Fig. 3b** and **3e**, those percentages would have been equally low had the tolerant animals been sacrificed earlier before 1 year posttransplant as Cohort B monkeys that had lost graft function. Nevertheless, as we do not know much about the kinetics of tolerance in different compartments in our model, we cannot rule out that the percentage of TEM cells in LMNCs and/or LNs in monkeys with long-term tolerance was higher before 1 year than at 1 year or later posttransplant. The revised Results section on **page 7, lines 22-28** addresses this point.

7. At the bottom of page 8, it reads, "...were 2 major biological pathways activated in sorted Tr1 cells in Cohort C, but Cohort B RM (Supplementary Table 7)". There seems to be a word missing in the segment I underlined.

Response:

Thank you for catching that oversight. We have added the missing word "not in" in the revised manuscript. (**Page 10, line 3**).

8. In the segment on Sensitization interferes with the tolerogenic efficacy of ADLs, there is discussion on how ADLs cultured with PBLs from sensitized animals stimulate greater proliferation of donor-specific CD4 and CD8 T cells. Previously, the authors stated that the stimulation resulted in abortive expansion with in vivo data showing PD1+ cells that may be anergic. Was any analysis performed on the donor-specific cells in sensitized animals to see if they were behaving differently and were not exhausted/anergic but activated? Otherwise it is hard to draw conclusions from this section.

Response:

To address this relevant point, we analyzed our previously generated phenotype FCS files to compare all available data on the PD-1 expression on circulating T cell subsets in PBLs in sensitized Cohort E monkeys (n=3) with the PD-1 expression in nonsensitized Cohort C

monkeys (n=5). Interestingly, analyses of circulating T cells at days +14 and at 1 month posttransplant revealed an average 2-fold increase in the percentage of circulating PD-1 expressing CD3+ (2.32 ± 0.21), CD4+ (1.94 ± 0.47), and CD8+ (2.12 ± 0.45) T cell subsets in Cohort C compared to Cohort E recipients (new **Supplementary Fig 15. a-c**). These observations suggest that T cells of recipients that were sensitized to donor at baseline remained activated after ADL administration. (**Page 11, lines 17-25**).

9. Supplementary Figure 3: The legend (solid red squares and solid blue circles) is inconsistent with the figure (solid blue squares and open red circles).

Response:

Thank you for pointing out that error, which we corrected in the revised manuscript.

Reviewer #3 (Remarks to the Author):

The manuscript by Singh and colleagues demonstrates the feasibility of inducing donor-specific tolerance to MHC class I mismatched, 1 DRB-class II-matched islet allografts in non-human primates with STZ-induced diabetes, without having to co-transplant bone marrow or induce donor chimerism. The authors use infusions of apoptotic donor leukocytes (ADL), in a regimen that could have easier clinical applicability than the regimens trying to induced mixed chimerism currently in clinical trials. However, the authors' regimen is less effective in full mismatched islet transplantation or in sensitized hosts. Mechanistically, they argue that ADL induces a regulatory network that suppresses residual donor-specific effector T cells.

The long-term allograft acceptance in the class I mismatch, 1 DRB-matched islets is quite impressive, with evidence of true donor-specific tolerance which is remarkable; the analysis of donor-specific responses using pMHC tetramers and functional readouts in vitro is elegant; and the characterization of the Tr1 cells and siRNA manipulation is convincing to infer the induction of regulatory mechanism. However, some issues limit enthusiasm for the current manuscript.

1. The authors ascribe their results to the use of apoptotic donor leukocytes (ADL) and they have extensive experience with the tolerogenic potential of ECDI-fixed cells, from mouse models of autoimmunity and transplantation to clinical trials in multiple sclerosis. However, in this particular study, the monkeys in the control group (cohort B) received transplantation

without immunosuppression, whereas the experimental group (cohort C) received ADL 2x plus blocking anti-CD40 plus rapamycin plus sTNFR and plus anti-IL-6R. The fact that 2/5 monkeys in the untreated control group accepted their allografts long-term implies that the immunogenicity of this donor/recipient combination is modest. Therefore, it is not possible to determine if the effects observed with the immunosuppressive treatment are in fact due to the ADL infusion, or to the other components of the immunosuppressive regimen, or to the combination of both, unless the authors have another control cohort that receives everything except ADL (and perhaps a control cohort that receives ADL alone, though this one is less crucial). This reviewer appreciates the cost of additional cohorts, but the authors cannot make the claims they make without them.

Response:

We perhaps could have been clearer in the description of the Cohorts, but we have done precisely what Reviewer #3 is proposing. The two main Cohorts studied in our manuscript are Cohorts B and C. Both Cohorts received the very same transient immunosuppression including anti-CD40 mAb 2C10, rapamycin, sTNFR, and anti-IL-6R mAb but only Cohort C received apoptotic donor leukocyte infusions in addition to immunosuppression.

The manuscript details in several sections that all recipient monkeys in Cohorts A-E were immunosuppressed, including the Cohort B animals. For more details, please see the paragraph titled "ADLs promote stable islet allograft tolerance in 1 DRB-matched RMs" in the Results section, as well as the paragraph on Study Animals and the **Supplementary Table 1**, and the paragraph on "Transient Immunosuppression", which are part of Supplementary Materials.

2. As a continuation of the previous point, if the full immunosuppressive regimen had been more effective in cohorts D (full mismatch) and E (sensitized), it may not have mattered as much to tease out the role of ADL versus the other drugs, as one might imagine moving the whole combo forward to the clinic. But because donor-specific tolerance is only achieved in the weaker combination, such that the regimen will need to be further optimized for more immunogenic donor/recipient combinations, it is important to understand the additional impact of ADL over that of the other biologics.

Response:

Please see also our response to the Reviewer's previous question. In light of the clarification that Cohort B recipients - as all other recipients studied in this manuscript - received transient immunosuppression, we feel that is fair to assume that the rejection documented in 5 of 7 of Cohort B recipients suggests that the donor-recipient combinations studied in this manuscript are not necessarily too weak to support eventual clinical translation of the findings.

3. The authors seem to overinterpret some of their findings. In particular, they claim that ADL induces Bregs. To the knowledge of this reviewer, there still isn't a marker that faithfully identifies Bregs in the way FoxP3 can identify Tregs. The authors use CD24⁺CD38⁺ as their definition of Bregs, but this likely only identifies activated B cells. It is possible that they contain a subset of Bregs, but it seems over interpreted to call these Bregs. Similarly for B10 cells and natural suppressor cells. Although it is possible, as the authors state, that ADL induces a regulatory network of Tregs, Tr1s, Bregs, MDCs etc, the evidence here, except for the Tr1 cells, is weak.

Response:

We agree with Reviewer #3 that the markers used to identify Bregs, B10 cells, NS cell, and other immune regulatory cells are less accurate than FoxP3 is in identifying Tregs also realizing that FoxP3 alone cannot be used to identify human Tregs.

To address this limitation, we have added the following statement to the Discussion (**page 15, lines 15-20**):

“Several lines of evidence suggest that CD24^{hi}CD38^{hi} Breg cells and CD24^{hi} CD27⁺ B10 cells are IL-10 producing B cells with potent regulatory function in autoimmunity, infection, and transplantation (Blair, Norena et al. 2010, Flores-Borja, Bosma et al. 2013, Jin, Weiqian et al. 2013, Nie, Wu et al. 2016, Qiu, Liu et al. 2017, Elizondo, Andargie et al. 2019, Liu, Qiu et al. 2019) . Our ex vivo data suggests that Breg cells suppress the proliferation of donor-specific T cells. Nevertheless, further studies are needed to verify the regulatory function of individual regulatory subsets expanded by ADL infusions as well as their interaction.”

References (New references discussed above and included in the revised manuscript)

Agashe, V. V., E. Jankowska-Gan, M. Keller, J. A. Sullivan, L. D. Haynes, J. F. Kernien, J. R. Torrealba, D. Roenneburg, M. Dart, M. Colonna, D. S. Wilkes and W. J. Burlingham (2018). "Leukocyte-Associated Ig-like Receptor 1 Inhibits Th1 Responses but Is Required for Natural and Induced Monocyte-Dependent Th17 Responses." J Immunol **201**(2): 772-781.

Blair, P. A., L. Y. Norena, F. Flores-Borja, D. J. Rawlings, D. A. Isenberg, M. R. Ehrenstein and C. Mauri (2010). "CD19(+)CD24(hi)CD38(hi) B cells exhibit regulatory capacity in healthy individuals but are functionally impaired in systemic Lupus Erythematosus patients." Immunity **32**(1): 129-140.

Chung, E. Y., J. Liu, Y. Homma, Y. Zhang, A. Brendolan, M. Saggese, J. Han, R. Silverstein, L. Selleri and X. Ma (2007). "Interleukin-10 expression in macrophages during phagocytosis of apoptotic cells is mediated by homeodomain proteins Pbx1 and Prep-1." Immunity **27**(6): 952-964.

Elizondo, D. M., T. E. Andargie, N. L. Haddock, R. L. L. da Silva, T. R. de Moura and M. W. Lipscomb (2019). "IL-10 producing CD8(+) CD122(+) PD-1(+) regulatory T cells are expanded by dendritic cells silenced for Allograft Inflammatory Factor-1." J Leukoc Biol **105**(1): 123-130.

Fabregat, A., S. Jupe, L. Matthews, K. Sidiropoulos, M. Gillespie, P. Garapati, R. Haw, B. Jassal, F. Korninger, B. May, M. Milacic, C. D. Roca, K. Rothfels, C. Sevilla, V. Shamovsky, S. Shorsler, T. Varusai, G. Viteri, J. Weiser, G. Wu, L. Stein, H. Hermjakob and P. D'Eustachio (2018). "The Reactome Pathway Knowledgebase." Nucleic Acids Res **46**(D1): D649-D655.

Flores-Borja, F., A. Bosma, D. Ng, V. Reddy, M. R. Ehrenstein, D. A. Isenberg and C. Mauri (2013). "CD19+CD24hiCD38hi B cells maintain regulatory T cells while limiting TH1 and TH17 differentiation." Sci Transl Med **5**(173): 173ra123.

Getts, D. R., D. M. Turley, C. E. Smith, C. T. Harp, D. McCarthy, E. M. Feeney, M. T. Getts, A. J. Martin, X. Luo, R. L. Terry, N. J. King and S. D. Miller (2011). "Tolerance induced by apoptotic antigen-coupled leukocytes is induced by PD-L1+ and IL-10-producing splenic macrophages and maintained by T regulatory cells." J Immunol **187**(5): 2405-2417.

Gregori, S., D. Tomasoni, V. Pacciani, M. Scirpoli, M. Battaglia, C. F. Magnani, E. Hauben and M. G. Roncarolo (2010). "Differentiation of type 1 T regulatory cells (Tr1) by tolerogenic DC-10 requires the IL-10-dependent ILT4/HLA-G pathway." Blood **116**(6): 935-944.

Jin, L., C. Weiqian and Y. Lihuan (2013). "Peripheral CD24^{hi} CD27⁺ CD19⁺ B cells subset as a potential biomarker in naive systemic lupus erythematosus." Int J Rheum Dis **16**(6): 698-708.

Jordan, S. C., J. Choi, I. Kim, G. Wu, M. Toyoda, B. Shin and A. Vo (2017). "Interleukin-6, A Cytokine Critical to Mediation of Inflammation, Autoimmunity and Allograft Rejection: Therapeutic Implications of IL-6 Receptor Blockade." Transplantation **101**(1): 32-44.

Kwan, T., S. J. Chadban, J. Ma, S. Bao, S. I. Alexander and H. Wu (2015). "IL-17 deficiency attenuates allograft injury and prolongs survival in a murine model of fully MHC-mismatched renal allograft transplantation." Am J Transplant **15**(6): 1555-1567.

Liu, H., F. Qiu, Y. Wang, Q. Zeng, C. Liu, Y. Chen, C. L. Liang, Q. Zhang, L. Han and Z. Dai (2019). "CD8⁺CD122⁺PD-1⁺ Tregs Synergize With Costimulatory Blockade of CD40/CD154, but Not B7/CD28, to Prolong Murine Allograft Survival." Front Immunol **10**: 306.

Marino, J., J. Paster and G. Benichou (2016). "Allorecognition by T Lymphocytes and Allograft Rejection." Front Immunol **7**: 582.

Matignon, M., A. Aissat, F. Canoui-Poitaine, C. Grondin, C. Pilon, D. Desvaux, D. Saadoun, Q. Barathon, M. Garrido, V. Audard, P. Remy, P. Lang, J. Cohen and P. Grimbert (2015). "Th-17 Alloimmune Responses in Renal Allograft Biopsies From Recipients of Kidney Transplants Using Extended Criteria Donors During Acute T Cell-Mediated Rejection." Am J Transplant **15**(10): 2718-2725.

Nie, L., W. Wu, Z. Lu, G. Zhu and J. Liu (2016). "CXCR3 May Help Regulate the Inflammatory Response in Acute Lung Injury via a Pathway Modulated by IL-10 Secreted by CD8⁺ CD122⁺ Regulatory T Cells." Inflammation **39**(2): 526-533.

Qiu, F., H. Liu, C. L. Liang, G. D. Nie and Z. Dai (2017). "A New Immunosuppressive Molecule Emodin Induces both CD4(+)FoxP3(+) and CD8(+)CD122(+) Regulatory T Cells and Suppresses Murine Allograft Rejection." Front Immunol **8**: 1519.

Roncarolo, M. G., S. Gregori, R. Bacchetta and M. Battaglia (2014). "Tr1 cells and the counter-regulation of immunity: natural mechanisms and therapeutic applications." Curr Top Microbiol Immunol **380**: 39-68.

Roncarolo, M. G., S. Gregori, R. Bacchetta, M. Battaglia and N. Gagliani (2018). "The Biology of T Regulatory Type 1 Cells and Their Therapeutic Application in Immune-Mediated Diseases." Immunity **49**(6): 1004-1019.

Uehara, M., Z. Solhjou, N. Banouni, V. Kasinath, Y. Xiaqun, L. Dai, O. Yilmam, M. Yilmaz, T. Ichimura, P. Fiorina, P. N. Martins, S. Ohori, I. Guleria, O. H. Maarouf, S. G. Tullius, M. M. McGrath and R. Abdi (2018). "Ischemia augments alloimmune injury through IL-6-driven CD4(+) alloreactivity." Sci Rep **8**(1): 2461.

Valujskikh, A. and P. S. Heeger (2000). "CD4+ T cells responsive through the indirect pathway can mediate skin graft rejection in the absence of interferon-gamma." Transplantation **69**(5): 1016-1019.

REVIEWERS' COMMENTS:

Reviewer #1 (Remarks to the Author):

The authors well responded to the questions and comments raised by the reviewer. One typo to be corrected: repetition of same words at p14, line 11 "proliferation of proliferation of".

Reviewer #2 (Remarks to the Author):

The authors did a fine job of addressing my comments and added new/additional information as necessary. I read the other Reviewer's comments with interest and felt that the authors also addressed their excellent critiques sufficiently. However, the addition of new information brought out one new comment and I have a few minor suggestions based on the new data/text.

1: The authors originally stated:

Previous murine studies showed that uptake of apoptotic bodies by APCs following ADL infusions substantially increased PD-L1/2 expression while downregulating positive costimulation¹¹. APCs exhibiting such patterns rapidly (but transiently) activated T cells that produce IFN-g and IL-10 but not IL-2, IL-6, and TNF-a^{35, 36}, a cytokine microenvironment known to promote apoptotic depletion of antigen-specific T cells^{37, 38}.

And they now added:

Several studies have demonstrated that donor-specific alloimmune responses can be mediated through the production of cytokines other than IFN-g including IL-6^{40, 41} and IL-17⁴²⁻⁴⁴ and that CD4⁺ T cells with indirect specificity can mediate skin graft rejection in the absence of IFN-g^{19, 45}. This begs the question of whether they analyzed the production of Th17 cytokines in their monkey cohorts (as the first statement was extracted from rodent data) and if the infusion of ADLs altered their production/expression. Since they administered anti-IL6R mAb, it would be an interesting point to discuss.

Minor comments:

Please remove the word "new" from "new Supplementary Fig. 15a-c", as that figure will not be new to a different reader. Also, capitalize the word "cohort" in the next sentence for consistency throughout the paper.

The sentence "Analysis of donor specific proliferation showed significant difference in proliferation of proliferation of CD4⁺ and CD8⁺ T cells at 1 month and 3 months posttransplant (Fig 3c,f)" would benefit from revision as it uses the word proliferation three times.

Reviewer #3 (Remarks to the Author):

The authors have addressed the concerns.

Reviewer #1

We wanted to thank Reviewer #1 for the review and the feedback.

Comment 1: One typo to be corrected: repetition of same words at p14, line 11 "proliferation of proliferation of".

Answer: Thank you for your comment. We have corrected the oversight on page 14.

Reviewer #2

We appreciate the comments provided by Reviewer #2.

Comment 1: The authors originally stated:

Previous murine studies showed that uptake of apoptotic bodies by APCs following ADL infusions substantially increased PD-L1/2 expression while downregulating positive costimulation¹¹. APCs exhibiting such patterns rapidly (but transiently) activated T cells that produce IFN-g and IL-10 but not IL-2, IL-6, and TNF-a^{35, 36}, a cytokine microenvironment known to promote apoptotic depletion of antigen-specific T cells^{37, 38}.

And they now added:

Several studies have demonstrated that donor-specific alloimmune responses can be mediated through the production of cytokines other than IFN-g including IL-6^{40, 41} and IL-17⁴²⁻⁴⁴ and that CD4⁺ T cells with indirect specificity can mediate skin graft rejection in the absence of IFN-g^{19, 45}

This begs the question of whether they analyzed the production of Th17 cytokines in their monkey cohorts (as the first statement was extracted from rodent data) and if the infusion of ADLs altered their production/expression. Since they administered anti-IL6R mAb, it would be an interesting point to discuss.

Answer: Thank you for your insightful critique. We have added new data on IL-17 protein levels measured in supernatants of MLR cultures of irradiated donor PBLs and Cohort B vs Cohort C recipient PBLs collected at intervals posttransplant. The following findings have been added to the 'Results' section" (page 8, lines 7-10): "*ADL infusions significantly suppressed IL-17 protein levels in supernatants of donor-stimulated PBLs collected at intervals posttransplant from Cohort C recipients when compared with IL-17 levels in posttransplant Cohort B MLRs (Supplementary Fig. 8j)*". Furthermore, the following text has been added to the 'Discussion' (page 14): "*Though IL-17 levels in posttransplant MLR cultures were suppressed in Cohort C but not in Cohort B, ...*". Finally, we added a description of the new experiment to the 'Methods' section (page 21).

Minor comments:

Comment 2: Please remove the word "new" from "new Supplementary Fig. 15a-c", as that figure will not be new to a different reader. Also, capitalize the word "cohort" in the next sentence for consistency throughout the paper.

Answer: The word "new" in "new Supplementary Fig. 15a-c" has been removed and "cohort" has been capitalized for consistency throughout the paper.

Comment 3: The sentence "Analysis of donor specific proliferation showed significant difference in proliferation of proliferation of CD4⁺ and CD8⁺ T cells at 1 month and 3 months posttransplant (Fig 3c,f)" would benefit from revision as it uses the word proliferation three times.

Answer: Thanks you for your comment. The sentence has been modified to "*Analysis of donor-specific proliferation showed significant difference in CD4⁺ and CD8⁺ T cell responses between Cohorts B and C at 1 month and 3 months posttransplant (Fig 3c,f)*".

Reviewer #3

No comments.

We wanted to thank Reviewer #3 for the review and the feedback.